# LLM-Check: Investigating Detection of Hallucinations in Large Language Models

**Gaurang Sriramanan**
gaurangs@cs.umd.edu

**Siddhant Bharti**
sbharti@cs.umd.edu

**Vinu Sankar Sadasivan**
vinu@cs.umd.edu

**Shoumik Saha**
smksaha@cs.umd.edu

**Priyatham Kattakinda**
pkattaki@umd.edu

**Soheil Feizi**
sfeizi@cs.umd.edu

Department of Computer Science
University of Maryland, College Park, USA

## Abstract

While Large Language Models (LLMs) have become immensely popular due to their outstanding performance on a broad range of tasks, these models are prone to producing hallucinations— outputs that are fallacious or fabricated yet often appear plausible or tenable at a glance. In this paper, we conduct a comprehensive investigation into the nature of hallucinations within LLMs and furthermore explore effective techniques for detecting such inaccuracies in various real-world settings. Prior approaches to detect hallucinations in LLM outputs, such as consistency checks or retrieval-based methods, typically assume access to multiple model responses or large databases. These techniques, however, tend to be computationally expensive in practice, thereby limiting their applicability to real-time analysis. In contrast, in this work, we seek to identify hallucinations within a single response in both white-box and black-box settings by analyzing the internal hidden states, attention maps, and output prediction probabilities of an auxiliary LLM. In addition, we also study hallucination detection in scenarios where ground-truth references are also available, such as in the setting of Retrieval-Augmented Generation (RAG). We demonstrate that the proposed detection methods are extremely compute-efficient, with speedups of up to 45x and 450x over other baselines, while achieving significant improvements in detection performance over diverse datasets.

## 1 Introduction

Over the past few years, Large Language Models (LLMs) such as GPT-4 [Achiam et al., 2023] and Llama [Touvron et al., 2023] have become immensely popular due to their excellent performance on natural language inference, question-answering and summarization tasks. Nonetheless, it has been observed that these models often produce outputs that are fallacious, incorporating fictional or insubstantial details that can be partly misleading or entirely fabricated [Ji et al., 2023, Zhang et al., 2023]. Moreover, it is often observed that such model generations seem plausible, appearing tenable before further scrutiny. This phenomenon in LLMs, known as hallucinations, poses a significant challenge to their deployment in applications where accuracy and reliability are critical. Thus, the detection and mitigation of hallucinations in LLMs have been subjects of critical interest. Furthermore, the relative difficulty of hallucination detection can vary significantly in white-box or black-box settings, depending on the extent to which access is available to the original LLM that was utilized for generation.

38th Conference on Neural Information Processing Systems (NeurIPS 2024).

Uncertainty estimation based on output logit based metrics such as perplexity or logit entropy have been used towards analyzing errors in structured prediction and generations produced by language models [Malinin and Gales, 2021, Kuhn et al., 2023]. Furthermore, a more fine-grained approach can be used by considering the relevance of specific tokens in the generated text [Duan et al., 2023], or by directly obtaining quality or score assessments leveraging the zero-shot capabilities of LLMs [Fu et al., 2023], depending upon the task specifications. As an alternative approach, Wang et al. [2023] showed that self-consistency can be incorporated into the decoding strategy to improve upon naive greedy-decoding in the setting of chain-of-thought prompting, and demonstrate notable improvements in arithmetic and common-sense reasoning benchmarks. Specific to the setting of hallucination detection, consistency-based methods like SelfCheckGPT [Manakul et al., 2023] and INSIDE [Chen et al., 2024] proposed to analyze multiple responses of the same model given a common prompt. Indeed, SelfCheckGPT [Manakul et al., 2023] relies upon the fact that when LLMs are trained on a given concept, its output generations tend to be more consistent, while hallucinated information is less likely to be repeatedly generated across multiple stochastically sampled responses. Notably, SelfCheckGPT relies only upon black-box based API access to models such as GPT-3.5 [Brown et al., 2020] to assess hallucinations in a given sample sentence. Similarly, Chen et al. [2024] proposed the INSIDE detection method based on eigenvalue analysis of the covariance matrix over model embeddings generated across multiple responses of the same prompt, enabling hallucination detection at a population level instead of sample-level detection. Another setting studied towards the detection of hallucinations is that of methods using Retrieval Augmented Generation (RAG) [Lewis et al., 2021, Guo et al., 2022, Varshney et al., 2023, Niu et al., 2024], which assume access to a large external database of grounded information, pipelined with a fact-verification system.

A key limitation of these detectors is the assumption that they have access to multiple model generations or a large reference dataset. Consistency-based methods assume hallucinations are rare, and most generations are factually correct, even though a given LLM model often repeats similar errors. Moreover, these techniques induce additional overheads to generate multiple additional responses at inference time, or require large memory overheads to parallelize generations. Retrieval-based techniques rely on strong fact-verification and retrieval techniques, and may not be scalable for large databases due to the considerable inference costs involved overall. Furthermore, incorporating human-level annotation or manual fact checking is often not scalable in this setting.

This leads us to our research question — Is it possible to identify the presence of hallucinations within a single LLM response in both white-box and black-box settings without incurring additional computational overheads at training or inference time? Towards this, given the practical constraints involved in this problem setting, we seek to analyze the efficacy of hallucination detection using analyses of the internal attention kernel maps, hidden activations and output prediction probabilities of an LLM itself, using an auxiliary substitute model if white-box access is unavailable. Furthermore, we study the effectiveness of different hallucination detection metrics both in the zero-knowledge setting as well as RAG-based settings where ground-truth reference materials are assumed to be available at inference time. We demonstrate that the detection methods proposed are extremely efficient by using teacher-forcing without additional overheads, and is accompanied by significant gains in detection performance relative to baseline approaches. In summary, we make the following contributions[1] in this work:

- We analyze hallucination detection within a single LLM response using its corresponding internal attention kernel maps, hidden activations and output prediction probabilities. We utilize these diversified scoring methods from different model components to potentially maximize the capture of hallucinations amongst its various forms without incurring computational overheads at training or inference time.
- We analyze the problem in broad-ranging settings across diverse datasets: from zero-resource detection (FAVA [Mishra et al., 2024]), to the case when multiple model responses are available (SelfCheck [Manakul et al., 2023]), or when ground-truth references are indeed available at inference time (RAGTruth [Niu et al., 2024]).
- We demonstrate that the proposed method LLM-Check is indeed effective over these diverse hallucination detection settings, and achieves notable improvements over existing baselines.
- Furthermore, we demonstrate such performance gains while requiring only a fraction of the computational cost (speedups up to 45x and 450x), as the proposed method does not utilize multiple model generations or extensive external databases.

---

[1]The codebase for LLM-Check is available at this URL

## 2 Related Works

Manakul et al. [2023] introduced SelfCheckGPT, a suite of hallucination detection methods in a zero-resource gray-box/black-box setting to assess the veracity of a LLM response, assuming only access to the output probability distributions. This is highly practical in real-world usage, since API calls often impede access to internal model activations. On the other hand, INSIDE [Chen et al., 2024] measures the self-consistency of the hidden states across multiple independent responses of an LLM, by performing a centered eigen-analysis of the covariance matrix of these hidden states. Notably, INSIDE follows a slightly different evaluation framework from other techniques which generally rely upon human annotations, wherein towards assessing the correctness of a model response INSIDE utilizes either ROUGE-L score or BERT-similarity scores being larger than a threshold with respect to a ground-truth. However, ROUGE-L is an n-gram based method that evaluates the longest common subsequence between the model response and the ground-truth. Thus, several hallucinatory words can be incorporated without a considerable change in the ROUGE-L score overall.

Kadavath et al. [2022] proposed to use the LLM itself to predict the probability of the generated response being True, referred to as Self-Prompt; this is primarily relevant for the context of multiple-choice-questions where the model explicitly has all the options in context. Furthermore, they also train models to predict the probability that a given multiple-choice-question will be correctly answered by calibrating its internal confidence values. Azaria and Mitchell [2023] proposed to train feedforward neural networks on the hidden activations of intermediate layers of LLMs on a True-False dataset of relatively short, simple sentences such as "The zebra uses flying for locomotion." This method is thus a white-box detection technique which also requires supervised training data to train the feedforward network (as noted by Manakul et al. [2023]); in contrast, we do not incorporate training of any sub-network or the LLM itself. Furthermore, it is fairly non-trivial to extend the technique towards more general settings such as when external references or multiple model responses are available at inference time. Yuksekgonul et al. [2024] investigate factual errors made by LLMs by modeling it as a constraint-satisfaction problem. Indeed, the factual query is specified by considering a sequence of Constraint-tokens and using a subsequent Verifier (such as ExactMatch or WikiData Search) to determine the satisfaction for each such constraint. This method is highly pertinent in settings where a fact-checking service is available for ready oracle-access that can provide feedback and annotations in real-time.

Mishra et al. [2024] introduced a fine-grained hallucination detection dataset (FAVA-Bench), wherein a novel hierarchy of hallucination types is identified and analyzed comprehensively. The authors first generate a synthetic hallucination training dataset by inserting specific errors using ChatGPT, and then fine-tune a Llama-2 model (FAVA) to explicitly detect these fine-grained hallucinations. The work also introduces a human-annotated dataset where the specific hallucination types are recorded with specified hallucination spans. We make extensive use of the FAVA datasets so released to compare multiple baselines on a common test-bed. Though retrieval augmented generation (RAG) helps to mitigate hallucination in LLMs, ungrounded generations still persist [Shuster et al., 2021, Li et al., 2024]. Recently, Wu et al. [2023] showed how popular LLMs still hallucinate on different tasks even with the integration of RAG, and compiled the RAGTruth dataset with span-level human annotation. Furthermore, they fine-tune an LLM on their training data to detect hallucinations and its span, but thereby incur a significant computational overhead at training time. Due to paucity of space, we extend our discussion of related works further in Section-D of the Appendix.

## 3 Taxonomy and Formalisms for Hallucination Detection

**Notation:** Let $\mathcal{V}$ denote the vocabulary of an LLM, and let $\mathcal{V}^*$ denote the set of all possible sequences obtainable using the same vocabulary. We denote an individual token as $x \in \mathcal{V}$, and a sequence of tokens as $\mathbf{x} \in \mathcal{V}^*$. For an autoregressive model $f$, let $p_f(\cdot|\mathbf{x})$ denote the next-token probability distribution based on input $\mathbf{x}$. Moreover, given two token sequences $\mathbf{x} = (x_1 x_2 \ldots x_n)$ and $\hat{\mathbf{x}} = (\hat{x}_1 \hat{x}_2 \ldots \hat{x}_m)$, let $\mathbf{x} \oplus \hat{\mathbf{x}} = (x_1 x_2 \ldots x_n \hat{x}_1 \hat{x}_2 \ldots \hat{x}_m)$ denote their ordered concatenation.

Given a query or prompt $\mathbf{x_p} = (x_1 x_2 \ldots x_n)$, let the LLM response of a model $f$ be denoted as $\mathbf{x} = (x_{n+1} \ldots x_m)$. Thus, the broad goal of detection is to determine the presence or absence of hallucination in the output response $\mathbf{x}$, when the input prompt is $\mathbf{x_p}$. In practice, LLMs such as chat-models require additional system prompts to be included along with the user input; here we assume these are contained within $\mathbf{x_p}$ for simplicity.

**Taxonomy of Hallucination Detection**

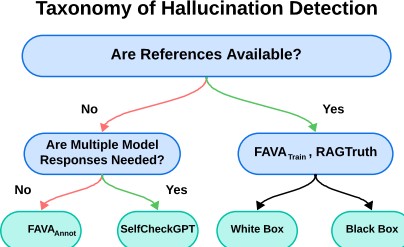

Figure 1: Taxonomy for different settings of hallucination detection, and the different datasets and data splits that correspond to each such setting.

| Method | Train Indep. | Single Response | Efficient | Sample Specific | Retrieval Indep. |
|---|---|---|---|---|---|
| FAVA | ✗ | ✓ | ✓ | ✓ | ✓ |
| SelfCheckGPT | ✓ | ✗ | ✗ | ✓ | ✓ |
| INSIDE | ✓ | ✗ | ✓ | ✗ | ✓ |
| RAGTruth | ✗ | ✓ | ✗ | ✓ | ✗ |
| LLM-Check (ours) | ✓ | ✓ | ✓ | ✓ | ✓ |

Table 1: Overview of Hallucination Detection Methods with FAVA, SelfCheckGPT, INSIDE, and RAGTruth. Key- Train Indep: does not require model fine-tuning/training, Single Response: does not require multiple LLM responses, Efficient: compute & memory efficient, Sample Specific: sample-wise instead of population-level detection, Retrieval Indep: does not require RAG or reference databases.

We present a broad taxonomy of the different settings in which hallucination detection is studied in Figure-1. Here, we discuss and analyze this classification in detail:

**I. Without External References:** First, we consider the setting wherein no external references are available to provide additional context to the LLM. Within this scenario, we can make a further classification based on whether multiple model responses can be potentially generated for the same prompt:

**IA. Single Model Response:** Here, only the original prompt $x_p$ and fixed model output $x$ is assumed to be available. Thus, in this setting, hallucination detection using LLMs relies upon the tacit assumption that the query topic is covered to some reasonable extent in the pre-training data of the LLM. We comprehensively analyze this setting using the FAVA Human-Annotated dataset to compare different approaches. Moreover, Mishra et al. [2024] provide a fine-tuned Llama-2-chat model, that can be used to annotate specific error types in the given response $x$. In our evaluations, we binarize these outputs to only detect the presence or absence of hallucinations.

**IB. Multiple Model Responses:** Given a prompt $x_p$ and a fixed model output $x$, SelfCheckGPT additionally generates stochastically sampled responses $x^1, x^2, \ldots x^k$ in order to compute different metrics such as BertScore, n-gram, Question-Answer and LLM-prompt score with respect to $x$, to assess the absence or presence of hallucinations in $x$. Notably, these scores can be computed with a black-box model with only external API access. On the other hand, INSIDE only considers the input prompt $x_p$, and computes eigen-decomposition of the covariance matrix of hidden activations across multiple generated samples $x^1, x^2, \ldots x^k$. This is used to then statistically infer the presence of hallucinations within this set of generations, based on a possible lack of self-consistency at a "population level" between samples within this specific set of $k$ samples. Thus, INSIDE performs hallucination detection at a population-level of all the generated samples simultaneously, rather than detection on a given fixed (single) output response $x$.

**II. With External References:** In this setting, a broad-ranging set of data retrieved from external sources are used to affix additional context to assess the validity of the model response $x$. This is a highly pertinent setting, as the retrieved data can consist of up-to-date information from sources deemed reliable or trustworthy. Furthermore, with the inclusion of external data, LLMs can be used on topics not completely covered in its original pretraining data in a straightforward manner, without requiring computationally intensive rounds of additional training or fine-tuning.

**IIA. White-Box:** For a given prompt $x_p$, we consider hallucination detection in an output response $x$ that is generated by a given LLM $f$. If the original LLM $f$ is available and accessible to compute the internal model activations, the setting is considered to be "white-box". For instance, INSIDE provides the population-level detection using the original source model so used, and thus falls in the family of white-box detection techniques.

**IIA. Black-Box:** If the original LLM $f$ that generated the response $x$ for prompt $x_p$ is no longer available or inaccessible, an auxiliary substitute LLM $\hat{f}$ can be utilized (such as open-source LLMs like Llama-2) to compute scores with internal activations and attention kernel maps. This can be achieved by using teacher-forcing on the substitute LLM, and this setting is considered to be "black-box".

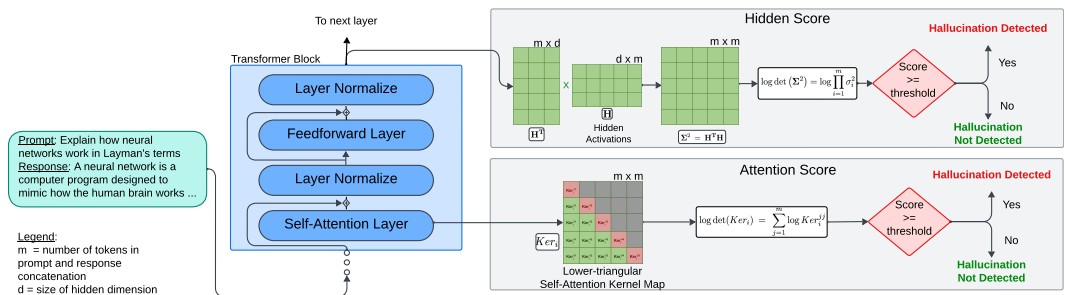

Figure 2: Schematic of detection pipeline using Eigenvalue Analysis of internal LLM representations.

# 4 Proposed Method

In this section, we describe our method LLM-Check in detail. Given that we wish to perform hallucination detection without any training or inference time overheads, we propose to analyze all model-related latent and output observables available with a single forward-pass of the LLM. For a broad family of autoregressive models using the Transformer architecture [Vaswani et al., 2017], the input token embeddings are transformed to a sequence of hidden representations at each layer of the model. For an LLM with $L$ internal layers, let $\mathbf{H}_l$ denote the hidden representations at layer $l \in \{1, \ldots, L\}$. The hidden representation at layer $(l + 1)$ is then computed as follows:

$$\mathbf{H}_{l+1} = \mathbf{H}_l + \mathbf{A}_{l+1} + \mathbf{M}_{l+1}$$

where $\mathbf{A}_{l+1}$ and $\mathbf{M}_{l+1}$ denote the attention and MLP contributions respectively. Additional operations such as layer-normalization are commonly carried out, but we omit their details here to be concise. The attention contributions represent the updates to the representation of a token at a given layer by attending to all token representations of the previous layer. In particular, if $Q, K, V$ denote the Query, Key and Values in the attention mechanism, the attention contribution is given by computing the kernel similarity of Q and K, and masking the values V with this kernel, and is simplified to be written as:

$$\mathbf{A} = Ker(Q, K)V = Softmax\left(\frac{Q \cdot K^T}{\sqrt{d_k}}\right) V$$

where $d_k$ is the dimension of the keys and queries. Generally, LLMs utilize multi-headed attention, wherein the kernel maps are individually computed over each head and concatenated, and then cast back to the appropriate dimension using Output projection matrices before being aggregated together. We note importantly that for auto-regressive LLMs, the attention kernel maps are lower triangular for each attention head, since a given token can only attend to the internal model representations of previous tokens in the sequence.

While LLMs do hallucinate, they still have a significantly appreciable degree of world-knowledge grounded on truthful facts encountered in the training stage, which is reflected in the fact that hallucinations are absent in some of the autoregressively generated sample outputs when multiple model responses are considered for the same prompt. In this work, we propose to directly analyze the variations in model characteristics between truthful and hallucinated examples by examining the rich semantic representations present in internal LLM representations, since we observe that the LLM can often generate the truthful version, albeit with potentially lower frequency. Thus, we posit that the model is in fact sensitive to the presence of non-factual, hallucinated information in responses which would otherwise be truthful, based on grounded precepts encountered in its training and that this can then be efficiently leveraged to perform hallucination detection within a single model response itself.

## 1. Eigenvalue Analysis of Internal LLM Representations

The differences in model sensitivity to hallucinated or truthful information would reflect in the rich semantic representations present in the hidden activations and the pattern of attention maps across different token representations. To quantitatively capture these variations, we propose to analyze the cross-covariance for hidden representations, and the kernel similarity map of self-attention across different tokens, since these form the foremost salient characteristics of the LLM itself. Given that we wish to do hallucination detection in black-box settings as well, we utilize teacher-forcing to

obtain representations corresponding to $\mathbf{x_p} \oplus \mathbf{x}$, wherein system prompt tokens delineate the original prompt and model output response. Thus, for a given layer $l \in \{1, \ldots, L\}$, let $\mathbf{H}$ denote the hidden representations for input $\mathbf{x_p} \oplus \mathbf{x}$. If the token embedding dimension is $d$ and $\mathbf{x_p} \oplus \mathbf{x}$ consists of $m$ tokens, each hidden representation matrix $\mathbf{H}$ is of shape $(d \times m)$. To be highly compute-efficient and enable real-time detection, we distill a simple yet salient scalar quantity - the mean log-determinant - from these variations in hidden representations using eigen-analysis. Theoretically, this is well-motivated as the eigenvalues and singular values capture the interaction in latent space between the different token representations corresponding to hallucinated and truthful sample sequences. We can compute the $(m \times m)$ covariance matrix $\mathbf{\Sigma}^2$, and compute its log-determinant as follows:

$$\mathbf{\Sigma}^2 = \mathbf{H^T H} \quad , \quad \log \det \left( \mathbf{\Sigma}^2 \right) = \log \prod_{i=1}^{m} \sigma_i^2 = \sum_{i=1}^{m} log\, \sigma_i^2 = 2 \sum_{i=1}^{m} log\, \sigma_i$$

where $\sigma_i$ are the singular values of $\mathbf{H}$. In order to remove the explicit dependence on input-length, we propose to utilize the mean log-determinant of $\mathbf{\Sigma}^2$, that is, $\frac{2}{m} \sum_{i=1}^{m} log\, \sigma_i$, which we term as Hidden Score. Here, we note crucial differences from the approach of INSIDE, which computes the centered covariance matrix *across multiple model responses* to check self-consistency of the set of responses, and performs hallucination detection at the population level for a set of responses, rather than a given fixed model response $\mathbf{x}$. We repeat this procedure across hidden representations of different layers; please refer to Section-F of the Appendix for detailed layer-wise analysis.

As an alternative approach, we now move towards utilizing internal components of the attention mechanism, namely the kernel similarity maps used within self-attention heads, to analyze hallucinations. If the number of self-attention heads utilized at each layer in the LLM architecture is $a$, the attention maps can be represented as tensors of the shape $(a \times m \times m)$ for an input sequence $\mathbf{x_p} \oplus \mathbf{x}$ of length $m$. If $Ker_i$ represents the kernel similarity map corresponding to attention head $i$, $Ker_i$ is a lower-triangular square matrix of size $(m \times m)$ for each $i \in \{1, \ldots, a\}$. Since it is lower-triangular, the eigenvalues of $Ker_i$ are just the values on the principal matrix diagonal, since they are the roots of the characteristic equation $\det(Ker_i - \lambda I) = 0$. Furthermore, since LLMs use the Softmax kernel, all the eigenvalues are also provably non-negative. We can thus obtain the log-determinant of $Ker_i$ without using SVD or eigen-decomposition:

$$\log \det(Ker_i) = \sum_{j=1}^{m} \log Ker_i^{jj}$$

where $Ker_i^{jj}$ represents the $(j, j)$ diagonal entry of the square matrix $Ker_i$. Again, we obtain the mean log-determinant for each attention head, and aggregate them to obtain an alternative hallucination measure, which we call Attention Score. Here, we note that computing the Attention Score is extremely efficient, since it does not require any explicit eigen-decomposition. The overall pipeline used for hallucination detection using the Hidden Score and Attention Score is visualized in Figure-2. We further present a detailed illustrative example using the Attention Score in Section-C of the Appendix.

In black-box settings, we propose to utilize an auxiliary autoregressive LLM $\hat{f}$, where we use teacher-forcing to obtain the scores from the auxiliary LLM instead to perform detection. We also note that since the MLP component of a transformer block is applied to each position separately and identically [Vaswani et al., 2017], it does not capture cross-token interactions, and thus we do not consider this component for detection itself.

## 2. Output Token Uncertainty Quantification

Alongside the internal model representations, we now consider the probability distribution $p_f(\cdot | \mathbf{x})$ over the output response tokens as induced by the LLM. Given that LLMs are trained with next-token prediction, the probability distribution $p_f(\cdot | \mathbf{x})$ for a given token can be highly salient toward the relative choices available for completion. Indeed, specifically for factual sentences that are part of topics covered in the training dataset of an LLM, we expect that its completions are such that the model likelihood of the output sequence is high. On the other hand, if the sentences arise from a distribution far different from that encountered in the training regime, the model outputs are more likely to be ungrounded in nature due to the lack of specificity between different tokens in the vocabulary.

We thus present the output token-level analysis techniques for identifying hallucinations in LLM responses. We propose to analyze the use of perplexity of the response $\mathbf{x}$, given the prompt $\mathbf{x_p}$, as a

potential measure for detecting hallucinations:

$$\text{PPL}(\mathbf{x}) = \exp\left(-\frac{1}{m-n+1}\sum_{i=n}^{m}\log p_f\left(x_i|\mathbf{x_p} \oplus \mathbf{x}_{<i}\right)\right)$$

Furthermore, at each token position in the output response, we can consider the mean entropy of the probability distribution over all tokens (not just the token finally selected during autoregressive generation). We can further refine logit-entropy-based measures by considering only the top-k tokens rather than all possible tokens since the contribution from tokens of extremely low probability is likely noisy and non-salient. We refer to this as the Logit Entropy score, defined as

$$\text{LogitEnt}(\mathbf{x}, k) = -\frac{1}{m-n+1}\sum_{i=n}^{m}\sum_{j=1}^{k} p_f(x_i^j|\mathbf{x_p} \oplus \mathbf{x}_{<i})\log p_f(x_i^j|\mathbf{x_p} \oplus \mathbf{x}_{<i})$$

where $x_i^j$ is the token with $j^{th}$ highest probability at output position $i$, conditioned on the prefix $\mathbf{x_p} \oplus \mathbf{x}_{<i}$ However, given that the entire sentence may not be hallucinatory, the length-average scores may not be as salient as we desire to predict hallucinations accurately. Thus, in light of this, we consider the Windowed Logit Entropy score, which computes the logit entropy scores across overlapping token windows and returns the logit entropy of the window with the maximum score. This score is thereby sensitive to short sequences of hallucinatory material and is not diluted by sequence length normalization.

We propose these two distinct lines of analysis towards hallucination detection, which we collectively term as LLM-Check, in order to adequately capture the extremely heterogeneous and diversified forms of hallucination displayed by modern Large Language Models over different domains. Towards this, the Eigen-analysis of internal LLM representations helps highlight the consistent pattern of modifications to the hidden states and the model attention across different token representations in latent space when hallucinations are present in model responses as compared to truthful, grounded responses. On the other hand, the uncertainty quantification of the output tokens helps analyze hallucinations based on the likelihood assigned by the model on the actual tokens predicted at a specific point in the sequence generated auto-regressively.

Before analyzing the quantitative experimental results in the following section, we first present qualitative comparisons of the proposed method with the most pertinent baselines in Table-1. We present various trade-offs and advantages in the table such as to whether the method requires fine-tuning of an LLM, if it inherently requires multiple model responses to detect hallucinations, if the method is computationally efficient, if it performs detection on per-sample basis or at a population level, and whether the method is inherently dependent on retrieval during inference time. We observe that our proposed approach is indeed qualitatively advantageous in all of the aspects so mentioned.

## 5 Experimental Results

**Datasets and Detection Details:** To analyze the efficacy of the proposed hallucination detection measures and perform a comparative analysis with existing baselines, we utilize the taxonomy and datasets presented in Figure-1. For the setting of detection without external references with a single model response, we utilize the FAVA-Annotation [Mishra et al., 2024] dataset, wherein we deem passages containing any form of hallucination to be hallucinated overall. Furthermore, we utilize the fine-grained classification of different forms of hallucination as annotated in the FAVA dataset to analyze the efficacy of the detection measures across these different types. In the setting of detection without external references with multiple model responses, we utilize the SelfCheckGPT dataset [Manakul et al., 2023], consisting of 1908 sentence level annotated samples, alongside 20 stochastically generated responses from GPT 3 (text-davinci-003). Moreover, to fairly compare across different detection methods, we also augment the originally released version of the FAVA-Annotation dataset by generating 20 additional responses from GPT 3 using the original prompt set. By doing so, we are able to obtain new baselines for SelfCheckGPT and INSIDE on the FAVA-Annotation dataset using the multiple responses so generated.

For our evaluations in the setting of hallucination detection with external references assumed available, we primarily consider the RAGTruth dataset [Wu et al., 2023]. In detail, we use the Summarization subset which was created by prompting six different LLMs with CNN/Daily Mail dataset and recent news, resulting in more than $5K$ samples. We use the span-level human annotation

| Model | Measure | AUROC | Accuracy | TPR @ 5% FPR | F1 Score |
|---|---|---|---|---|---|
| Llama-2-7B | Self-Prompt | 50.30 | 50.30 | - | 66.53 |
| Llama-2-7B | FAVA Model | 53.29 | 53.29 | - | 43.88 |
| Llama-2-7B | SelfCheckGPT-Prompt | 50.08 | 54.19 | - | 67.24 |
| Llama-2-7B | INSIDE | 59.03 | 57.98 | 13.17 | 39.66 |
| LLM-Check (Ours) | | | | | |
| Llama-2-7B | PPL Score | 53.22 | 58.68 | 3.59 | 68.33 |
| | Window Entropy | 56.90 | 56.59 | 2.99 | 42.52 |
| | Logit Entropy | 53.80 | 55.99 | 2.99 | 56.73 |
| | Hidden Score (LY 20) | 58.44 | 58.08 | 11.98 | 59.66 |
| | Attn Score (LY 21) | **72.34** | **67.96** | **14.97** | **69.27** |
| Vicuna-7B | PPL Score | 53.96 | 56.89 | 3.59 | 64.20 |
| | Window Entropy | 55.24 | 58.38 | 5.99 | 66.02 |
| | Logit Entropy | 52.29 | 55.69 | 1.80 | 57.31 |
| | Hidden Score (LY 15) | 58.22 | 59.28 | 10.18 | **66.99** |
| | Attn Score (LY 19) | **71.69** | **66.47** | **24.55** | 62.00 |
| Llama-3-8B | PPL Score | 53.22 | 58.68 | 3.59 | 67.40 |
| | Window Entropy | 56.90 | 56.59 | 2.99 | 55.52 |
| | Logit Entropy | 53.80 | 55.99 | 2.99 | 56.27 |
| | Hidden Score (LY 15) | 57.10 | 57.78 | 10.78 | 65.38 |
| | Attn Score (LY 23) | **68.19** | **65.87** | **15.57** | **70.53** |

Table 2: Detection results on the FAVA-Annotation Dataset wherein no External References are available. For methods such as INSIDE and SelfCheckGPT-Prompt, we utilize the multiple model responses generated by GPT-3.

so provided in the dataset and considered a sample is hallucinated if there exists any hallucinated span in the response. Additionally, we also consider white-box and black-box detection settings for the RAGTruth dataset, depending on the original LLM model used for generating the responses. Apart from RAGTruth, we also consider the training data-split of FAVA wherein several references are provided, and specific hallucination errors are edited in synthetically to obtain pairs of ground-truth responses with and without hallucinations. We consider a 500-sample subset of the FAVA train-data towards this setting of detection with external references.

**Models and Metrics:** We utilize popular open-source LLM chat-models such as Llama-2-7b Chat [Touvron et al., 2023], Vicuna [Zheng et al., 2023] and Llama-3-instruct [AI@Meta, 2024] as our autoregressive LLMs with their corresponding tokenizers provided by HuggingFace [Wolf et al., 2020], for both white-box and black-box evaluations. We compute the suite of hallucination detection scores proposed in Section-4, which we collectively term as LLM-Check, and present standard threshold based detection metrics such as Accuracy, Area under the ROC curve (AUROC), True Positive Rate (TPR) at low (5%) False Positive Rate (FPR) and F1 score. In each setting, we consider balanced datasets, with an equal number of samples with and without hallucinations present, except for the SelfCheck dataset where we follow the setup utilized in the original paper.

## 5.1 Detection Results on Datasets with no External References as Context

First, we analyze the setting of hallucination detection when no grounded references are provided as context. We present the consolidated detection metrics in Table-2 using the FAVA-Annotation dataset. To evaluate methods such as SelfCheckGPT and INSIDE, we make use of the stochastically sampled responses of GPT-3 that we augment to the original dataset. We observe that in this zero-context setting, many of the baselines perform fairly poorly in comparison to LLM-Check. We remark that while entropy-based scores were used for comparisons in the SelfCheckGPT paper, the best-performing variant was seen to be SelfCheckGPT-Prompt using GPT-3 itself, which was used to generate the original hallucinations. Here, we observe that when using Llama2-7B for SelfCheckGPT-Prompt, the results are much worse. This might be due to the fact that SelfCheckGPT is less effective at the passage level after aggregation is performed over the individual sentences. We observe that the Attention Scores are the most effective across different models, and obtains significantly higher detection scores.

| Model | Method | AUC-PR | Accuracy | TPR @ 5% FPR |
|---|---|---|---|---|
| Llama-2 | SelfCheck | 72.84 | 51.44 | 4.81 |
| Llama-3 | SelfCheck | 75.06 | 54.84 | 5.10 |
| LLM-Check (Ours) | | | | |
| Llama-2 | Attn Score | **80.04** | 58.91 | 9.41 |
| Llama-2 | Prompt | 79.46 | **61.21** | 8.76 |
| Llama-3 | Attn Score | 79.96 | 58.92 | **9.48** |
| Llama-3 | Prompt | 78.49 | 58.54 | 7.11 |

Table 3: Detection results on the SelfCheckGPT Dataset wherein no External References are available, but when multiple model responses are indeed available. We report the results using the same data-split as the original SelfCheck [Manakul et al., 2023] paper, and thus the number of positive and negative samples are imbalanced in this table: 1392/1908 samples have hallucinations present.

We further analyze the layer-wise performance of scores obtained from internal model representations in Section-F of the Appendix. We observe that while the Attention Scores are generally the most effective, particularly around Layer 20, the oscillation in detection performance across different layers can be large, relative to the case of using the Hidden Score which is fairly more stable across layers.

In Table-3, we further analyze the performance of LLM-Check against SelfCheckGPT on the SelfCheckGPT Dataset, where multiple GPT-3 generated responses are available in addition to the original model response. We follow the setup used in the original paper with imbalanced classes, and similarly also report Area under the Precision-Recall curve in-place of AUROC. We again find that the Attention scores provide better detection performance over all three metrics of AUC-PR, Accuracy and TPR@5%FPR, despite not utilizing the additional model responses included in the dataset. Here, we present a Prompt-variant of LLM-Check, where instead of prompting the LLM to output "Yes" or "No" based on the additional model responses and then hard-coding to 0/1 scores as in SelfCheckGPT-Prompt, we compute Attention Scores of modified prompts which include the additionally generated model responses, and then aggregate over them as done in SelfCheckGPT-Prompt. We find that LLM-Check with Attention-Prompt performs similarly well, though the inference time is higher in this setting, due to the iteration over the different model responses.

## 5.2 Detection Results on Datasets with External References

We now analyze the setting wherein external references are assumed to be available to the model at inference time. In Table-4, we present evaluations obtained using a Llama-2-7b model in both white-box and black-box settings. For the black-box setting, we considered four different models – Llama-2-13b, Llama-2-70b, GPT-4, Mistral-7b. In the white-box setting, we observe that the Hidden Score achieves a higher F1 score compared to all other detection methods. The Attention scores are however better in the overall black-box setting, where we compute a weighted average since different models hallucinate at different frequencies on the same prompt data. Additionally, we also observe that LLM-Check performs better on larger models. For example, with Hidden score, our approach obtains $54.11\%$, $59.67\%$, and $59.31\%$ AUROC on the 7b, 13b, and 70b variants of Llama-2 respectively. Furthermore, the AUROC increases to $61.87\%$ for hallucinations arising from the GPT-4 model.

Lastly, we present the evaluations on 500 examples of the FAVA Train-split in Table-5. Notably, this dataset differs from RAGTruth in that the hallucinations in the generations are inserted synthetically using GPT-3, and thus potentially induce changes to the joint-distribution of the sequence level probabilities. Indeed, we observe that output-token uncertainty estimates using entropy out-perform even the attention-based scores in this setting. Thus, we observe that different detection methods may prove optimal depending on the problem setup and underlying data distribution.

## 5.3 Time-Complexity Analysis

We compare the overall runtime cost of the proposed detection scores with other baselines using a Llama-2-7b Chat model on the FAVA-Annotation dataset on a single Nvidia A5000 GPU in Figure-3. We observe that the Logit and Attention scores are indeed very efficient, while the Hidden Score is slightly slower since it uses SVD. We also observe that LLM-Check is considerably faster than most baselines with speedups of up to 45x and 450x, since it only uses model representations with teacher

| Target Model | Measure | White-box | Black-box | | | | |
|---|---|---|---|---|---|---|---|
| | | Llama-2-7b | Llama-2-13b | Llama-2-70b | GPT-4 | Mistral-7b | Overall |
| Hidden Score | AUROC | 54.11 | 59.67 | 59.31 | 61.87 | 53.68 | 57.24 |
| | Accuracy | 56.33 | 59.66 | 58.42 | 68.52 | 54.15 | 57.62 |
| | TPR@5%FPR | 8.14 | 12.41 | 9.9 | 3.7 | 5.18 | 8.37 |
| | F1 Score | 61.51 | 50.42 | 66.14 | 67.86 | 32.58 | 47.45 |
| Logit (Perplexity) | AUROC | 53.73 | 52.46 | 56.97 | 52.13 | 52.11 | 53.27 |
| | Accuracy | 54.07 | 55.17 | 57.92 | 59.26 | 54.66 | 55.79 |
| | TPR@5%FPR | 7.69 | 8.97 | 6.93 | 0.00 | 4.15 | 6.01 |
| | F1 Score | 58.7 | 50.57 | 61.26 | 61.02 | 43.23 | 50.45 |
| Logit (Win Entropy) | AUROC | 52.08 | 55.71 | 56.38 | 55.83 | 52.61 | 54.58 |
| | Accuracy | 53.17 | 56.9 | 57.43 | 59.26 | 53.89 | 55.90 |
| | TPR@5%FPR | 4.98 | 15.86 | 1.98 | 7.41 | 10.36 | 10.08 |
| | F1 Score | 53.98 | 33.68 | 62.01 | 54.9 | 49.29 | 47.51 |
| Logit (Log Entropy) | AUROC | 53.95 | 51.18 | 55.14 | 50.34 | 50.43 | 51.68 |
| | Accuracy | 55.43 | 53.79 | 57.43 | 57.41 | 53.89 | 54.83 |
| | TPR@5%FPR | 7.24 | 9.66 | 4.95 | 0.00 | 6.22 | 6.65 |
| | F1 Score | 53.74 | 15.09 | 66.41 | 60 | 48.41 | 42.62 |
| Attention Score | AUROC | 54.19 | 60.05 | 60.01 | 63.51 | 55.37 | 58.30 |
| | Accuracy | 54.52 | 59.66 | 60.89 | 66.67 | 56.99 | 59.23 |
| | TPR@5%FPR | 5.88 | 14.48 | 12.87 | 7.41 | 5.18 | 9.87 |
| | F1 Score | 54.5 | 55.97 | 55.06 | 67.8 | 57.72 | 57.18 |

Table 4: Detection on the RAGTruth Dataset using Llama-2-7b model in white-box and black-box setting, with the "Overall" column presenting the weighted average results for the black-box models.

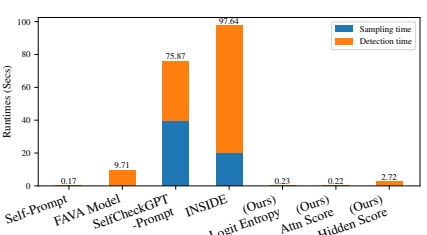

Figure 3: Averaged runtime analysis of different hallucination detection methods.

| Model | Measure | AUROC | Accuracy | TPR @ 5% FPR |
|---|---|---|---|---|
| Llama-2 | PPL Score | 74.20 | 70.00 | 26.00 |
| | Window Entropy | **77.00** | **72.00** | **34.00** |
| | Logit Entropy | 74.36 | 71.00 | 26.00 |
| | Hidden Score | 51.44 | 54.00 | 4.00 |
| | Attn Score | 69.57 | 66.60 | 11.60 |
| Llama-3 | PPL Score | 73.48 | 68.80 | 13.20 |
| | Window Entropy | 78.44 | 72.00 | **28.00** |
| | Logit Entropy | **79.24** | **73.60** | 28.00 |
| | Attn Score | 71.91 | 68.20 | 19.60 |

Table 5: Detection results on synthetic hallucinations on FAVA Train Split data with External References included.

forcing, without additional inference time overheads. This runtime analysis for the Eigenvalue based methods represents the **total time** for computing Attention and Hidden scores from **all 32 layers** of Llama-2-7b. That is, the Attention score computation for all 32 layers takes 0.22 seconds per example, and the Hidden score computation for all 32 layers requires 2.72 seconds per example, when averaged over the FAVA Annotation dataset. Since the overhead to compute scores for all layers is so small, we expect that they can be utilized for real-time detection of hallucinations in model responses.

## 6 Conclusions

In this work, we analyze the problem of detecting hallucinations within a single response of an LLM, and propose LLM-Check, an effective suite of techniques that only rely upon the internal hidden representations, attention similarity maps and logit outputs of an LLM. We demonstrate the efficacy of LLM-Check over broad-ranging settings and diverse datasets: from zero-resource detection to cases where multiple model generations or external databases are made available at inference time, or with varying access restrictions to the original source LLM. Moreover, we observe that LLM-Check obtains considerable improvements over baseline methods in these settings, without requiring fine-tuning or retraining of LLMs. Furthermore, by utilizing only teacher-forcing at inference time without additional overheads, we show that LLM-Check is extremely compute-efficient, requiring only a fraction of the runtime compared to other detection baselines, with speedups of up to 45x and 450x.

# 7 Acknowledgments

This project was supported in part by a grant from an NSF CAREER AWARD 1942230, ONR YIP award N00014-22-1-2271, ARO's Early Career Program Award 310902-00001, Army Grant No. W911NF2120076, the NSF award CCF2212458, NSF Award No. 2229885 (NSF Institute for Trustworthy AI in Law and Society, TRAILS), an Amazon Research Award and an award from Capital One.

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

# Appendix

## A   Limitations and Future Work

**1. Improving Hallucination Detection Performance:** In this work, we consider different detection methods using the internal representations and output probabilities of a well-trained LLM. We thus assume access to at least one proxy open-source LLM such as Llama to compute our detection scores, which may not always be feasible. This is however a mild assumption, as we do believe that this covers a very large fraction of practical use cases encountered in the real-world. Moreover, the overall detection rates have significant scope for further improvement, especially towards improving the True Positive Rate at low False Positive Rates.

**2. Mitigation of Hallucinations:** While this work focuses on detection of hallucinations, the mitigation of hallucinations in LLMs itself is not addressed directly. We hope that future works can possibly build upon the scoring metrics proposed here towards reducing the frequency of hallucinations, which would be a potentially significant contribution overall. We anticipate that LLM-Check could be directly incorporated towards providing additional automated feedback in the fine-tuning stage of LLMs with Reinforcement Learning, wherein output sample generations that are detected to be hallucinatory in nature can be down-weighted appropriately. Additionally, the detection methods can assist in flagging samples in a highly efficient manner towards a customized human-feedback loop with RLHF, wherein annotators can introduce an orthogonal ranking which reflects the desired extent of factuality for the sample generations so considered.

**3. Principled Incorporation of Retrieved Data:** Furthermore, we are also hopeful that future work towards incorporating external references in a more principled manner could help in this detection task using Retrieval Based Augmentation [18]. For instance, the proposed detection methods can perhaps serve as an initial screening mechanism: if potential hallucinations are detected, a subsequent RAG-based query can be made. This process would be triggered when the detector flags any indications of hallucination, allowing for a deeper, context-aware verification of the content produced by the LLM with minimized running costs. This two-step method would not only enhance the precision of the detection itself, but also integrates additional grounded verification steps to ensure the reliability and accuracy of the model outputs.

**4. Combining different Detection Techniques:** We attempt to combine the different scoring methods proposed for hallucination detection, namely using Eigen-analysis of internal LLM representations and Output Token Uncertainty Quantification methods, towards a unified form of detection. We observe that this problem can be reduced to that of learning a classifier that takes the different scores as input and is trained to predict the absence or presence of hallucinations. We conducted preliminary experiments towards this on the FAVA Annotation dataset using Logistic Regression, but observed no appreciable gains over using the best performing individual method, the Attention Score, on this dataset. We seem to observe that learning an optimal combination of the different scores in a manner that it generalizes well across diverse truthful and hallucinatory inputs is a highly non-trivial task, especially considering the different forms of hallucination highlighted in the fine-grained taxonomy presented in the FAVA dataset. In this paper, we focus on fairly simple combinations of these scores, given the strong motivation to keep the overall compute cost for detection very low to enable its use in real-time systems. However, we do hope that future works can build upon the proposed methods fruitfully to achieve significantly higher detection performance, albeit with a potentially higher computational footprint.

## B   Broader Impact

In this work, we propose LLM-Check that can be used for efficient hallucination detection in critical scenarios even when the ground-truth reference or multiple model responses might not be available. We believe our work is an important step toward reliable hallucination detection in these constrained scenarios. Our work could positively impact studying and mitigating hallucinations in LLMs. Similar to other detection methods in the literature, a false detection with our method could lead to some model generations being flagged as hallucinations or some hallucinated contents not being detected as hallucinated. However, in our experiments, we show that our method LLM-Check reduces these errors

when compared to other baseline hallucination detection techniques. To the best of our knowledge, direct negative societal impacts due to our work are minimal.

## C  Visualizing the Attention Scores

To inspect the variation across hallucinatory and non-hallucinatory examples, we utilize the FAVA Train split dataset, since it consists of pairs of samples with and without hallucinations for the same exact prompt. We make use of the following illustrative example, which post-tokenization contains the same total number of tokens for both the hallucinated and truthful responses. This helps simplify the comparison since the length-normalization is not an additional factor towards differentiating the scores. We list the prompt $\mathbf{x_p}$ (with the actual references shown shortened here for visualization purposes, since they are fairly long), alongside the hallucinated Sample (HS) and the Truthful sample (TS):

---

Prompt $\mathbf{x_p}$: Read the following references:
Reference [1]: The Ballad of Big Al ...
Reference [2]: ...
Reference [3]: ...
Reference [4]: ...
Reference [5]: ...

Based only on the references provided, write a blog article.

---

Hallucinated Sample (HS): "The Song of Big Al" is a special episode of the nature documentary series "Walking with Dinosaurs" that focuses on the life story of an Tyrannosaurus specimen called "Big Al". The story is based on a well-preserved fossil of Big Al, which lived during the Early Jurassic period approximately 145 million years ago. The episode was produced by the BBC Natural History Unit and partnered with the National Geographic Channel, ProSieben, and TV Asahi. Rumor has it that the episode was partially shot in Cresswell Craggs, UK. Additionally, a behind-the-scenes episode called "Big Al Uncovered" was aired alongside "The Song of Big Al"

Truthful Sample (TS): "The Ballad of Big Al" is a special episode of the nature documentary series "Walking with Dinosaurs" that focuses on the life story of an Allosaurus specimen called "Big Al". The story is based on a well-preserved fossil of Big Al, which lived during the Late Jurassic period approximately 145 million years ago. The episode was produced by the BBC Natural History Unit and partnered with the Discovery Channel, ProSieben, and TV Asahi. Rumor has it that the episode was partially shot in Cresswell Craggs, UK. Additionally, a behind-the-scenes episode called "Big Al Uncovered" was aired alongside "The Ballad of Big Al"

---

We observe that the overall response structure is quite similar, but key phrases such as "The Song of Big Al" is hallucinated as "The Ballad of Big Al", "the Discovery Channel" is hallucinated as "the National Geographic Channel", and "Tyrannosaurus specimen called "Big Al"" is hallucinated as "Allosaurus specimen called "Big Al"".

We then analyze the eigenvalues of the attention kernel using a Llama-2-7b model, which are used to compute the proposed Attention Score. We know from standard results that eigenvectors with distinct eigenvalues are orthogonal, and thus we focus on analyzing the eigenvalues directly, rather than the high-dimensional eigenvectors to illustrate and motivate the proposed method. We highlight the key differentiating tokens along with the log-eigenvalue corresponding to that specific token position in the attention mechanism at an intermediate layer which contributes to the total mean-log-determinant in the proposed Attention Score in Table-6:

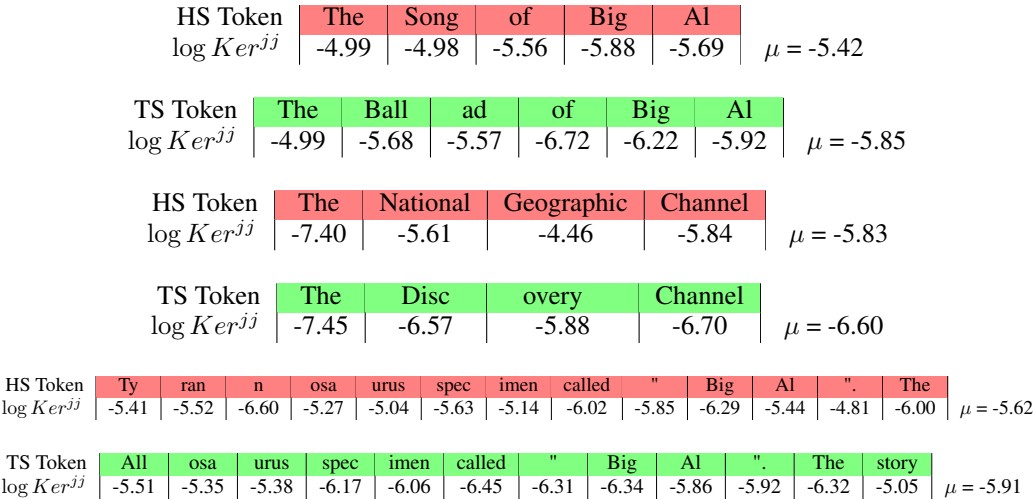

| HS Token | The | Song | of | Big | Al | |
|---|---|---|---|---|---|---|
| $\log Ker^{jj}$ | -4.99 | -4.98 | -5.56 | -5.88 | -5.69 | $\mu = -5.42$ |

| TS Token | The | Ball | ad | of | Big | Al | |
|---|---|---|---|---|---|---|---|
| $\log Ker^{jj}$ | -4.99 | -5.68 | -5.57 | -6.72 | -6.22 | -5.92 | $\mu = -5.85$ |

| HS Token | The | National | Geographic | Channel | |
|---|---|---|---|---|---|
| $\log Ker^{jj}$ | -7.40 | -5.61 | -4.46 | -5.84 | $\mu = -5.83$ |

| TS Token | The | Disc | overy | Channel | |
|---|---|---|---|---|---|
| $\log Ker^{jj}$ | -7.45 | -6.57 | -5.88 | -6.70 | $\mu = -6.60$ |

| HS Token | Ty | ran | n | osa | urus | spec | imen | called | " | Big | Al | ". | The | |
|---|---|---|---|---|---|---|---|---|---|---|---|---|---|---|
| $\log Ker^{jj}$ | -5.41 | -5.52 | -6.60 | -5.27 | -5.04 | -5.63 | -5.14 | -6.02 | -5.85 | -6.29 | -5.44 | -4.81 | -6.00 | $\mu = -5.62$ |

| TS Token | All | osa | urus | spec | imen | called | " | Big | Al | ". | The | story | |
|---|---|---|---|---|---|---|---|---|---|---|---|---|---|
| $\log Ker^{jj}$ | -5.51 | -5.35 | -5.38 | -6.17 | -6.06 | -6.45 | -6.31 | -6.34 | -5.86 | -5.92 | -6.32 | -5.05 | $\mu = -5.91$ |

Table 6: $\log Ker^{jj}$ computed over the diagonal of lower triangular attention kernels for substrings containing Hallucinated Sample (HS) tokens and Truthful Sample (TS) tokens, averaged across different attention heads. We observe that the overall mean $\mu$ over different tokens, is consistently larger for Hallucinated Sample Tokens.

We thus observe that the log-eigenvalues of the Hallucinated response are indeed larger in value, indicating that the rich latent space representations of the LLM are indeed indicative of the presence of hallucinated text. In the third example pair of hallucinated and truthful samples, we see that though the hallucinated details are split across more tokens than that in the truthful response, we observe that the LLM is sensitive to the error as the log-eigenvalues corresponding to the tokens that immediately follow the error are larger, again contributing to the overall detection that the overall response is indeed hallucinated.

In Figure-4, we also visualize the cumulative difference in the $\log Ker^{jj}$ values from the first token till the $j^{th}$ token (between 1 and total length) between the Hallucinated Sample (HS) and Truthful sample (TS). Though we observe that the difference in log-eigenvalues between the hallucinated and truthful responses is not entirely monotonic throughout the token sequence, we observe that the log-eigenvalues corresponding to the hallucinated response are consistently larger over the whole sequence when compared to the log-eigenvalues arising from the truthful response.

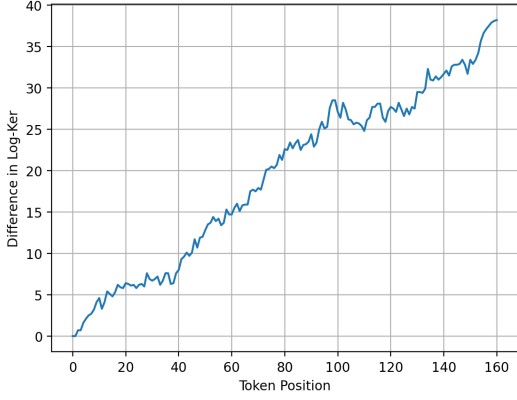

Figure 4: Difference between the cumulative sum of log-eigenvalues from the first token till the $j^{th}$ token (between 1 and total length) between the Hallucinated Sample (HS) and Truthful sample (TS).

We observe this phenomenon in greater generality, such as for hallucinated responses of different lengths compared to truthful ones. We highlight that this is a key advantage of using the mean-log-

determinant which normalizes using the length of the token sequence, as compared to scoring methods such as Negative-log-likelihood which do not explicitly account for varying sequence lengths.

## D Related Works (Continued)

In this section, we continue our discussion on related works due to paucity of space in the main paper. [9] study hallucinations specifically in the setup of Neural machine translation (NMT), and even propose mitigation techniques to overwrite hallucinatory sections of the output. [21] proposed to model predictive uncertainty by using a Dirichlet Prior Network to explicitly parameterize a distribution over distributions on a simplex. [15] proposed Semantic Entropy (SE), wherein multiple sequences are first generated, and then subsequently clustered based on bi-directional entailment again using a language model. After this equivalence relation between different responses is established using clusters, they can be treated as categorical datapoints, and thus the semantic entropy can be estimated. This technique is similar to Semantic Entropy based estimation for a single response which is performed in "SelfCheckGPT with BERTScore" and "SelfCheckGPT with NLI" in their original SelfCheck-GPT paper, but these are seen to perform worse than their best scoring method "SelfCheckGPT with Prompt". We note that LLM-Check performs better than SelfCheckGPT-Prompt as well (Tables-2,3). Furthermore, Semantic Entropy is very costly to estimate for real-time detection since the several model inference steps are required to establish the clusters using the equivalence relation proposed. Both Predictive and Semantic entropy quantify the total uncertainty over the distribution of all possible responses given a fixed prompt, in sharp contrast to the primary focus of this paper, which analyzes the hallucinatory behavior within a single fixed model response output on a given prompt. Thus, Predictive Entropy and Semantic Entropy perform population-level uncertainty estimation akin to INSIDE, and is not amenable to the single-response case.

Comparison with standard classifiers on hidden representations: [20] proposed Inference-Time Intervention (ITI), which improves the truthfullness of LLMs by shifting internal model activations along truth-correlated directions. To do this, the authors identify a sparse set of attention heads with high linear probing accuracy for truthfulness using a base dataset such as TruthfulQA, and subsequently perturb activations during autoregressive generation in inference time. We note that methods which rely upon supervised training of classifiers on internal model representations over samples with and without hallucinations can add large computational overheads. For instance, Inference-Time Intervention (ITI) relies upon training 1024 binary classifiers on the TruthQA datasets, and thus becomes prohibitively expensive (as also noted by [5]).

Nevertheless, we experiment with standard classifiers on hidden representations on the FAVA Annotation dataset. Namely, we create a train-test split of the dataset, and train classifiers on the hidden representations corresponding to layer 20 and layer 30 of a Llama-2-7B model. We use this split to ensure that we have about 100 testing samples balanced between classes (with and without hallucinations) to obtain reliable evaluations of the classifiers so trained. We observe that using layer 20, this method obtains AUROC, Accuracy and TPR@5%FPR of 55.76, 59.82 and 0.00 respectively, while using layer 30 we observe detection rates with AUROC, Accuracy and TPR@5%FPR of 56.52, 61.61 and 1.79 respectively. We note that despite requiring supervised training, this method performs significantly worse than the proposed Attention Score which achieves AUROC, Accuracy and TPR@5%FPR of 72.34, 67.96 and 14.97 respectively. We hypothesize that the generalization of these classifiers might not be adequate to achieve reliable detection performance across the diverse hallucination samples and types as present in the FAVA dataset, as the corresponding hidden representations become similarly disparate.

[29] utilizes the persistence homology dimension (PHD) from topological data analysis, specifically to differentiate the inherent dimensionality for real and AI-generated texts, distinct from the detection of hallucinated versus truthful, grounded responses. However, these techniques developed towards characterizing the intrinsic dimensionality of the underlying data manifold can plausibly be extended to differentiate between truthful and hallucinatory texts. [35] proposed to utilize a Distance-aware maximum likelihood estimation (MLE) for the Local Intrinsic Dimension (LID), by fitting to a Poisson distribution, where the rate of the Poisson process is parameterized by the intrinsic dimension, in order to determine the truthfulness of model responses. We do note key differences, such as that LID estimation requires explicit optimization (minimization of heteroskedastic weighted polynomial regression is performed), and utilizes the latent space representation corresponding to the last token alone, in sharp contrast to our proposed method which considers the latent representations

over the entire token sequence. Furthermore, this technique requires the availability of a large number of neighboring samples, such as 200 samples, to estimate LID effectively. Thus, this technique is more similar to INSIDE and standard classifiers on trained hidden representations, as compared to our work which focuses on detection in the single-response setting.

# E Experimental Results (Continued)

| Measure | AUROC | Accuracy | TPR @ 5% FPR |
|---|---|---|---|
| Entity Hallucinations (374 samples) | | | |
| PPL Score | 60.14 | 59.89 | 5.35 |
| Window Entropy | 57.35 | 58.02 | 2.67 |
| Logit Entropy | 45.91 | 52.14 | 3.21 |
| Hidden Score (Layer 20) | 67.93 | 65.78 | 14.44 |
| Attention Score (Layer 20) | 64.64 | 61.23 | 21.39 |
| Relation Hallucinations (50 samples) | | | |
| PPL Score | 58.08 | 62.00 | 4.00 |
| Window Entropy | 52.64 | 56.00 | 8.00 |
| Logit Entropy | 55.84 | 64.00 | 0.00 |
| Hidden Score (Layer 21) | 58.72 | 58.00 | 8.00 |
| Attention Score (Layer 21) | 64.32 | 66.00 | 4.00 |
| Invented Hallucinations (120 samples) | | | |
| PPL Score | 69.03 | 68.33 | 5.00 |
| Window Entropy | 62.25 | 61.67 | 3.33 |
| Logit Entropy | 44.08 | 52.50 | 0.00 |
| Hidden Score (Layer 20) | 80.53 | 75.00 | 16.67 |
| Attention Score (Layer 20) | 75.22 | 69.17 | 31.67 |
| Subjective Hallucinations (182 samples) | | | |
| PPL Score | 48.36 | 57.14 | 0.00 |
| Window Entropy | 54.76 | 56.59 | 1.10 |
| Logit Entropy | 61.15 | 62.09 | 4.40 |
| Hidden Score (Layer 15) | 56.32 | 60.99 | 8.79 |
| Attention Score (Layer 21) | 67.14 | 67.58 | 9.89 |
| Unverifiable Hallucinations (174 samples) | | | |
| PPL Score | 49.25 | 59.77 | 1.15 |
| Window Entropy | 54.95 | 57.47 | 3.45 |
| Logit Entropy | 60.22 | 63.22 | 2.30 |
| Hidden Score (Layer 20) | 63.32 | 63.22 | 12.64 |
| Attention Score (Layer 5) | 67.83 | 69.54 | 10.34 |

Table 7: Hallucination Detection without References on the FAVA Annotated Dataset: Detection metrics obtained on a balanced dataset with different forms of hallucinations as per the FAVA taxonomy.

Here, we analyze the effectiveness of different detector types on different forms of hallucinations as presented within the taxonomy introduced by FAVA. Thus, we consider samples that originate from one of the following types of hallucinations: "entity", "relation", "contradictory", "invented", "subjective" and "unverifiable".

We present these consolidated results in Table-7. Here, we again observe that the Attention Scores are quite effective in detecting hallucinations across different types. However, we do observe fairly large oscillations across layers in each of these settings, and the token-based measures might be more consistent, even though their absolute numbers are slightly lower.

In Figure 5, we show the ROC curves for logit-based detectors on the annotated FAVA dataset for entity and relation hallucinations. Note that the negative detector scores might help detect some kinds of hallucination instances. For example, while negative perplexity scores help in entity hallucination detection, positive perplexity scores help in relation hallucination detection. This implies that the detection scheme could be perhaps improved by using interval based detection, rather than threshold

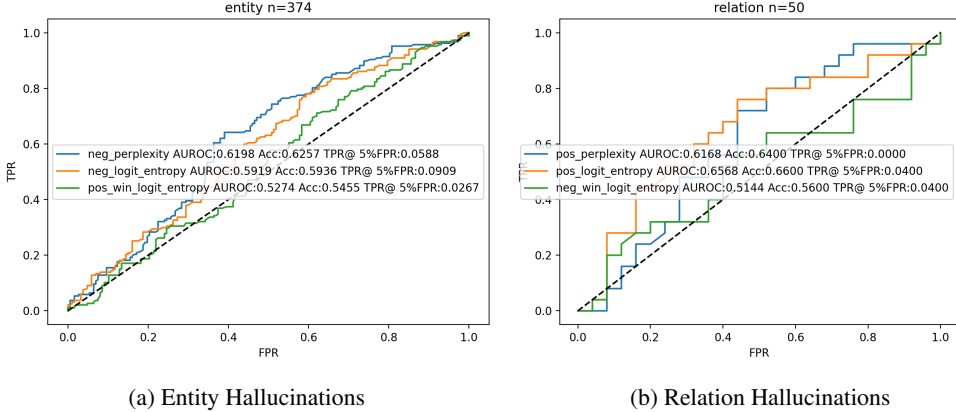

(a) Entity Hallucinations          (b) Relation Hallucinations

Figure 5: ROC curves for logit-based detection schemes with the annotated FAVA dataset. Figures 5a and 5b show the detection of entity and relation hallucinations, respectively. As observed here, taking the negative detection scores helps with detecting various kinds of hallucinations.

based detection. Thus, for the logit level detection scores, it is beneficial to consider both positive and negative scores to obtain large difference in detection performance.

## E.1 Experimental Details and Hyperparameter Choices

For all our evaluations, we use Pytorch [25] models, primarily open-source LLM models available on Huggingface [33]. By default, we set the generation configuration of the Huggingface model to be {"temperature": 0.6, "top_p": 0.9, "top_k": 50, "do_sample": True}. For the setting of Logit Entropy, we consider the top 50 tokens as the selected candidates to compute the score. For the computation of Hidden scores, we do not observe a noticeable difference between using the covariance matrix of hidden representations, and a centered covariance matrix (as used in INSIDE). For the Hidden scores and Attention scores, we report the best of results obtained over all the layers, varying between 1 and 32 for Llama-2-7b and Vicuna-7b. For the FAVA Annotated dataset, we utilize a balanced subset consisting of 167 pairs of samples with and without hallucinations present. On the FAVA train set, we evaluate our results on the first 250 examples, which again consist of pairs of samples with and without hallucination, so to obtain 500 total samples. For the SelfCheck dataset, we utilize all 1908 annotated sentences provided. For detecting hallucinations in a given sentence, an increasing passage context is utilized by incorporating prior sentences arising from the same model response, since detection of hallucinations within a single sentence without the appropriate context will be ill-defined. Moreover, the model is prompted using "This is a Wikipedia passage about {concept}:" to specify the concept within WikiBio. This is not required in the case of consistency based methods, since the other reference samples explicitly provide this concept as is.

## E.2 Robustness Across Domains

In this paper, we attempt to cover an adequately diverse extent of domains between the FAVA-Annotation dataset, FAVA-train split, SelfCheckGPT Dataset and RAGTruth dataset. Indeed, the FAVA Annotation dataset itself spans four different data sources: Knowledge-intensive queries sampled from the Open Assistant dataset [14], Open QA prompts from the No Robots dataset [26], Instruction-following and WebNLG [8] datasets. In addition, the FAVA-train split consists of Wikipedia articles and Question Answering datasets [16], while SelfCheckGPT Dataset consists of Annotations from the WikiBio dataset [17]. Lastly, the RAGTruth dataset consists of annotations from the CNN/Daily Mail dataset [11]. To further assess robustness across sub-domains, we utilize the FAVA Annotation dataset, since the individual samples are explicitly labeled with the original dataset that they were derived from. Namely, FAVA Annotation dataset lists the following three named datasets: Open Assistant, Instruction-following and WebNLG. We compute the mean attention scores for each of these data-subsets, and observe the following values: -4.97, -4.94 and -5.12 over the Instruction-following dataset, Open Assistant dataset and WebNLG dataset respectively. We thus

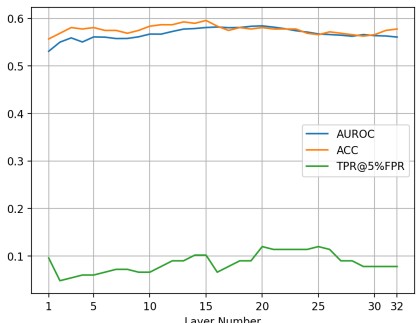 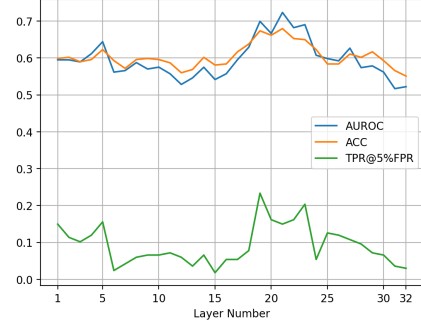

Figure 6: Layer-wise Analysis obtained with (a) Hidden Score and (b) Attention Score, on the left and right respectively for a Llama-7B model on the FAVA-Annotation dataset. Here, we can observe that the Attention based scores tend to yield higher AUROC, Accuracy and TPR@5%FPR, but can oscillate more strongly across layers. Moreover, we point out that Layer 19 using Attention Score obtains the highest TPR@5%FPR of 23.35%, though its AUROC is slightly lower than Layer 21.

observe that the scoring method is indeed consistent across different domains, and expect such results to continue to hold over subdomains of other commonly used datasets.

### E.3 Performance Differences compared to results reported by Prior Baselines

While the SelfCheckGPT paper shows that SelfCheckGPT-Prompt is the strongest baseline in their comparisons versus other approaches, we observe a extreme reduction in performance when we perform our evaluations with Llama-2 for SelfCheckGPT-Prompt, which is also the default model used in the released SelfCheck Github repository in the whitebox non-API setting. We posit that the SelfCheckGPT-Prompt results might have been potentially biased due to the fact that the multiple responses were indeed generated by GPT-3 (text-davinci-003) over the WikiBio dataset, and similar models were then used for the final evaluation. Moreover, it does appear that output uncertainty estimation based detectors, though generally weaker than the Hidden or Attention scores, can be salient detectors such as in the case with synthetic hallucinations in the FAVA-train data split. Furthermore, the exact entropy based scores considered in this paper are slightly more sophisticated than the ones considered in the SelfCheckGPT paper.

## F   Layerwise Analysis for Eigenvalue based Scores

### F.1   Layerwise Analysis and Layer Selection

As noted in Section-5.3, we observe that the method is computationally efficient even when all model layers are used for computing Hidden or Attention scores. Indeed, the runtime analysis shown in Figure-3, represents the total runtime for computing Attention scores and Hidden scores from **all** 32 layers of Llama-2-7b. That is, the Attention score computation for all 32 layers takes 0.22 seconds per example, while the Hidden score computation for all 32 layers requires 2.72 seconds per example when averaged over the FAVA Annotation dataset. Since the overhead to compute scores for all layers is so small, we expect that they can be utilized for real-time analysis.

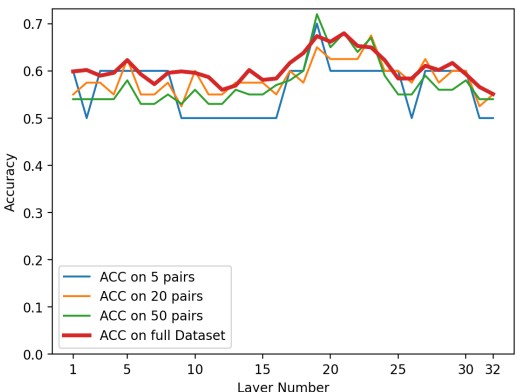

Figure 7: Results across different layers of Llama-2-7B obtained using sample subsets of 5, 20 and 50 pairs, as well as the complete dataset. We observe that general layerwise trends begin to hold with fairly limited sample pairs, and can help choose an optimal layer in an efficient manner.

In this section, we investigate methods towards layer selection for the eigenvalue based scoring methods with low computational overhead. From Figure-6, we first observe that the performance obtained with the Hidden score is extremely stable across layers, and thus it is relatively easy to choose, though we recommend a middle-level layer such as layer 20 for a 32 layer, 7-billion parameter model such as Llama-2. On the other hand, we do indeed observe a larger degree of oscillations across layers with the Attention score. Here, we perform an experiment to potentially rapidly select layers, by plotting the results obtained using few samples, and subsequently check if the overall performance on the dataset can be estimated using this for each layer.

We present these evaluations in Figure-7. We do indeed observe a fair degree of agreement between results obtained with 5, 20 and 50 pairs as compared to the full dataset. In general, we do observe that the layers between 19 and 23 achieve close-to-optimal performance for the Attention score computed on a 32 layer, 7-billion parameter model such as Llama-2. We hypothesize that for the white-box setting, while the very early layers are involved in feature extraction, and the last layers are involved more towards making an optimal next-token prediction, the layers after the midpoint of the network are quite suitable for hallucination detection. In the black-box setting, this is much more difficult since it is non-trivial to map representations of intermediate layers between different LLMs, especially when the original LLM has many more layers, as with GPT-4 or Llama-2-70B. Furthermore, the optimal layer can depend on factors such as the structure of knowledge presented within an example. For instance, samples within the SelfCheck dataset were created by performing a sentence-level split of paragraphs generated by GPT-3 based on WikiBio data, with corresponding sentence-level annotations. However, for detecting hallucinations in a given sentence, an increasing passage context is utilized by incorporating prior sentences arising from the same model response since detection of hallucinations within a single sentence without the appropriate context is not well-defined. Thus, in these cases of increasing context for sentence level detection, even early-layers are observed to be optimal, as compared to the case seen with other datasets such as FAVA-Train, FAVA-Annotation or RagTruth which are more standardized in their structure.

## F.2 Layerwise Detection Results on the RAGTruth Dataset

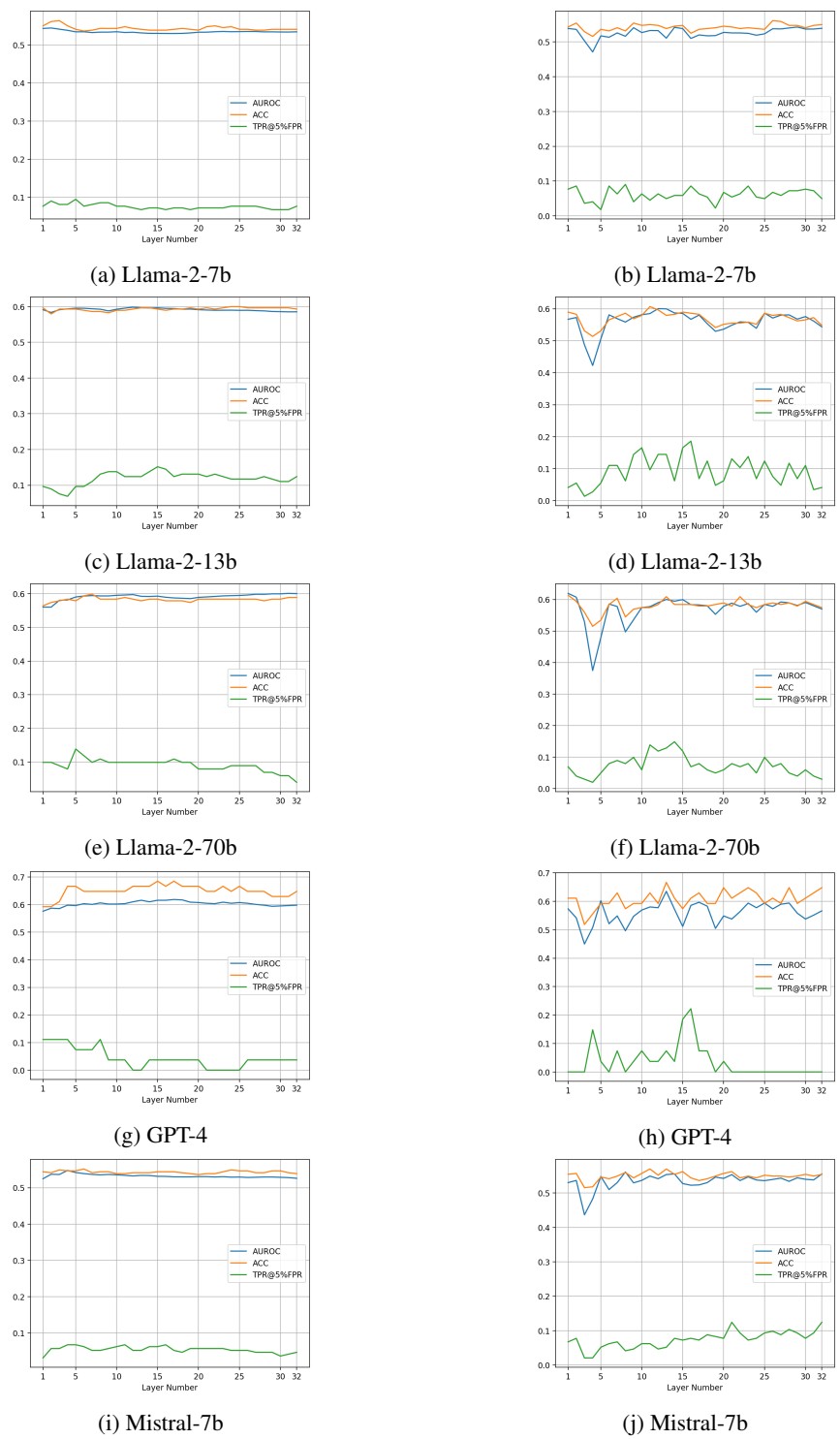

Figure 8: Performance for different layers with Hidden Score (left column) and Attention Score (right) on the RAGTruth dataset. All these scores are with the Llama-2-7b model.

