# OpenReview forum: "LLM-Check: Investigating Detection of Hallucinations in Large Language Models"
_NeurIPS.cc/2024/Conference — NeurIPS 2024 poster_

### Official Review · Reviewer_C5LP · 2024-07-07

**Soundness:** 2
**Presentation:** 2
**Contribution:** 2
**Rating:** 5
**Confidence:** 4

**Summary:**

The paper addresses the issue of hallucinations in large language models (LLMs). They propose using special scores based on internal model states such as attention maps, hidden activations, and output prediction probabilities to identify hallucinations within a single response in both white-box and black-box settings. Additionally, they explore scenarios where ground-truth references are available, such as in Retrieval-Augmented Generation (RAG). The proposed methods demonstrate significant improvements in detection performance while being computationally efficient.

**Strengths:**

1. The details for reproducibility are included as well as the code.
2. The paper covers a wide range of scenarios for hallucination detection, including settings with and without external references and varying levels of access to the model (white-box vs. black-box).
3. The paper includes extensive empirical evaluations using different datasets

**Weaknesses:**

1. Lack of theoretical background. The paper uses scores from other papers with different setups, while not explaining why this scores should work. For example, it is not clear why EigenScore method from INSIDE [1], specifically designed to measure self-consistency across different generations, will work on hidden activations and attention maps. Authors eigenscore on attention maps are just some modification of the trace of the matrix, but maybe the original trace is better?
2. Some important comparisons are missing. Authors compare with other techniques, but they don't provide any comparisons with standard usage of hidden activations in previous works for hallucination detection i.e [2, 3].
3. Domain robustness is not assessed. Score-based and classifier-based models for hallucination detection may not be robust for different text domains, but the paper misses this part.

[1] C. Chen, K. Liu, Z. Chen, Y. Gu, Y. Wu, M. Tao, Z. Fu, and J. Ye. INSIDE: LLMs’ internal states retain the power of hallucination detection. In The Twelfth International Conference on Learning Representations, 2024

[2] Amos Azaria, & Tom Mitchell (2023). The Internal State of an LLM Knows When It's Lying. In The 2023 Conference on Empirical Methods in Natural Language Processing.

[3] Kenneth Li, Oam Patel, Fernanda Viegas, Hanspeter Pfister, & Martin Wattenberg (2023). Inference-Time Intervention: Eliciting Truthful Answers from a Language Model. In Thirty-seventh Conference on Neural Information Processing Systems.

**Questions:**

1. How theoretically could be explained the use of EigenScore in this setup (not for self-consistency across generations)?
2. Could you provide comparisons with standard classifiers on hidden representations (if possible)?
3. Could you please explain more carefully about MLP contributions in lines 177-178?

Some minor typos:

Line 90 typo “an centered”

Line 138 ambiguous statement “in the response x, in the context of xp”

**Limitations:**

1. Mitigation of hallucination itself is not addressed, which is justified
2. The proposed methods are applicable only to response-level detection
3. Authors admit that using their model for calculating scores could be infeasible in real-world scenarios

---

> ### Author Rebuttal · Authors · 2024-08-07
>
> We thank the reviewer for their valuable feedback. We are encouraged that the reviewer appreciates the efficacy of the proposed method extensively demonstrated across diverse detection settings and datasets. We respond to the questions raised below:
>
>
> **Theoretical insights and comparisons with INSIDE**
> - In this work, we study the highly challenging task of hallucination detection within a single example - of the form $(x_p \oplus x)$ for a prompt $x_p$ and corresponding model response $x$ - without any training or inference time overheads. This sharply contrasts with INSIDE which performs population-level detection by computing the  centered covariance matrix across multiple model responses to check self-consistency of the set so generated.
> - Indeed, though the mean-log-determinant formula appears similar to that of INSIDE, in LLM-Check the interaction in latent space between the different token representations **within the same sample response** is utilized, as we posit that the LLM is in fact sensitive to the presence of non-factual, hallucinated information in responses which would otherwise be truthful. This arises from the probabilistic language-modeling learnt by the LLM, based on factual precepts encountered in its training and that this can then be efficiently leveraged to perform hallucination detection within a single model response itself.
> - Theoretically, this is well-motivated as the eigenvalues and singular values capture the changes in hidden interactions and patterns to the extent of attention applied, which we know is different in hallucinated samples which contain non-truths compared to non-hallucinated sample sequences. We also remark that the Attention score can be found without explicitly computing the SVD or eigen-decompositions since it is lower-triangular, enabling speedups between 44x and 450x compared to existing baselines, while also simultaneously achieving significantly improved detection performance.
> - Furthermore, the attention values represent softmax activations of the given attention head, and thus represent probability values arising from the cross-variation of the Keys and Values in the Attention Mechanism. Thus, the log-determinant correctly reduces to the trace of the log-values and characterizes the joint distribution of the predicted probabilities in the attention head, while the trace of the values themselves represents the **sum** of distinct probability values which is not theoretically sound, as these are not upper-bounded by 1.
>
> **Single-Response Analysis:** As noted by the reviewer, in the work we do perform hallucination detection within a single prompt-output response. However, we strongly believe that this is an *inherent advantage and not a limitation*, as the single-response analysis can be extended to population-level analysis by statistically aggregating scores across multiple output responses. We crucially note that this is strictly unidirectional - in that population-level detection methods such as INSIDE cannot be easily applied for single-response analysis. Moreover, we believe that the single-response analysis is more relevant for real-time detection of hallucinations in practical real-world use-cases.
>
>
>
> **Robustness across Domains:** In the paper, we do attempt to cover some extent of diversity in domains between the FAVA-annotation dataset, FAVA-train split,  SelfCheckGPT Dataset and RAGTruth dataset. Indeed, the FAVA Annotation dataset itself spans four different data sources: Knowledge-intensive queries sampled from the Open Assistant dataset (Kopf et al. , 2023), Open QA prompts from the No Robots dataset (Rajani et al., 2023), Instruction-following and WebNLG (Gardent et al., 2017) datasets. In addition, the FAVA-train split consists of Wikipedia articles and Question Answering datasets (Kwiatkowski et al., 2019), while SelfCheckGPT Dataset consists of Annotations from the WikiBio dataset (Lebret et al., 2016). Lastly, the RAGTruth dataset consists of annotations from the CNN/Daily Mail dataset (See et al., 2017).
> We do however agree that this could be further expanded to more domains like medical datasets, but the annotation process specifically for hallucinations can be very expensive in such cases since it often requires a great deal of expertise to peruse and annotate carefully. However, we do believe that if specific domains such as a medical-data is considered, wherein a LLM specifically fine-tuned on such data is used, hallucination detection techniques such as LLM-Check will continue to be highly effective.
>
> **Real-world Feasibility:** We wish to clarify that we do indeed believe that the proposed method is feasible in real-world scenarios, and relies upon the very mild assumption that at least one open-source LLM such as Llama is available to serve as a proxy to compute our detection scores. While this assumption may not always strictly hold in all scenarios, we do believe that this covers a very large fraction of practical use cases encountered in the real-world. Furthermore, the effective black-box performance indicates a considerable real-world advantage over methods such as INSIDE which is strictly white-box and requires multiple output responses at inference time.

---

> ### Author Response · Authors · 2024-08-07
> **Rebuttal Continued**
>
> *Comparison with standard classifiers on hidden representations:* We thank the reviewer for the suggestion. We first note that methods that rely upon supervised training of classifiers on internal model representations over samples with and without hallucinations, can add large computational overheads. For instance, Inference-Time Intervention (ITI) [2] relies upon training 1024 binary classifiers on the TruthQA datasets, and thus becomes prohibitively expensive as also noted by Chen et al. in the INSIDE paper.
> - However, as suggested by the reviewer, we experimented with standard classifiers on hidden representations on the FAVA Annotation dataset. Namely, we create a train-test split of the dataset, and train classifiers on the hidden representations corresponding to layer 20 and layer 30 of a Llama-2-7B model. We use this split to ensure that we have about 100 testing samples balanced between classes (with and without hallucinations) to obtain reliable evaluations of the classifiers so trained.
> - We observe that using layer 20, this method obtains AUROC, Accuracy and TPR@5%FPR of 55.76, 59.82 and 0.00 respectively, while using layer 30 we observe detection rates with AUROC, Accuracy and TPR@5%FPR of 56.52, 61.61 and 1.79 respectively. We note that despite requiring supervised training, this method performs significantly worse than the proposed Attention Score which achieves AUROC, Accuracy and TPR@5%FPR of 72.34, 67.96 and 14.97 respectively.
> - We hypothesize that the generalization of these classifiers might not be adequate to achieve reliable detection performance across the diverse hallucination samples and types as present in the FAVA dataset as the corresponding hidden representations become similarly disparate.
>
>
>
>
>
> *MLP Contributions in Transformer Block:* The MLP contribution is the standard form in the transformer block at layer $ l $ is given by
> $M_l$  = $W^1_l $ $\rho ($ $W^2_l ($  $A_l$ + $H_{l-1}))$, as the residual block incorporates both the attention kernel of the same layer and hidden representation of the previous layer, as introduced by Vaswani et al. in the original transformers paper. We note that since the MLP is applied to each position separately and identically, it does not capture cross-token interactions, and thus we do not consider this component for detection itself.
>
>
>
>
> *Additional Comments:* We also thank the reviewer for pointing out the typo in line 90, we will correct this error in the final version of the paper.
> In Line 137-138, “to determine the presence or absence of hallucination in the response $x$, in the context of $x_p$” refers to the detection of hallucinations in the output response $x$ when the input prompt is $x_p$.
>
>
>   We thank the reviewer for the detailed suggestions and constructive comments. We kindly ask if the reviewer would consider increasing their score if their concerns or questions have been addressed. We would be glad to engage further during the discussion period.

---

> ### Comment · Reviewer_C5LP · 2024-08-08
>
> Thank you for the detailed and comprehensive answer. I could agree with some remarks, especially about Single-Response Analysis as an advantage. Also, I would like to highlight the comparison provided with standard classifier looks convincing. However, I would also point out that some of remarks are not properly addressed possibly due to misunderstanding. I will clarify my concerns:
>
> **Theoretical motivation**: In INSIDE this method was well motivated, exactly because EigenScore measured the interactions between different samples. The method works because similar samples would give co-aligned eigenvectors and low EigenScore. In this work, you measure the same score but for different tokens' embeddings, and I believe it is not evident why should they be more aligned in non-hallucinatory content. I think the method is efficient, but I miss the motivation or explanation, that would definitely strengthen your study. As a suggestion, you could try to inspect eigenvectors and eigenvalues for special examples of hallucinatory and non-hallucinatory content.
> Moreover, this metric seems to be related to linear intrinsic dimensionality of embeddings. The intrinsic dimensionality of embeddings was used, for example, in [1] or [2].
>
> **Robustness across Domain**: when commenting about robustness to domain, I mean that if scores are robust to domain, the distribution of scores should be comparable for Wikipedia articles and Question Answering datasets from FAVA. Could you provide some results proving that scores are consistent across different distributions of texts?
>
> I will increase my score, if you could discuss my concerns and provide some results.
>
> [1] Tulchinskii, Eduard, et al. "Intrinsic dimension estimation for robust detection of ai-generated texts." Advances in Neural Information Processing Systems 36 (2024).
>
> [2] Yin, Fan, Jayanth Srinivasa, and Kai-Wei Chang. "Characterizing truthfulness in large language model generations with local intrinsic dimension." arXiv preprint arXiv:2402.18048 (2024).

---

> > ### Author Response · Authors · 2024-08-12
> > **Rebuttal Discussion (1)**
> >
> > **Motivating the Proposed Method:** We sincerely thank the reviewer for the clarifications and suggestions. To inspect the variation across hallucinatory and non-hallucinatory examples, we utilize the FAVA Train split dataset, since it consists of pairs of samples which do not contain or do contain hallucinations for the same exact prompt. We make use of the following illustrative example, which post-tokenization contains the same total number of tokens for both the hallucinated and truthful responses. This helps simplify the comparison since the length-normalization is not an additional factor towards differentiating the scores. We first list the hallucinated response (HR) and the Truthful response (TR):
> >
> > Hallucinated Response (HR): *"The Song of Big Al" is a special episode of the nature documentary series "Walking with Dinosaurs" that focuses on the life story of an Tyrannosaurus specimen called "Big Al". The story is based on a well-preserved fossil of Big Al, which lived during the Early Jurassic period approximately 145 million years ago. The episode was produced by the BBC Natural History Unit and partnered with the National Geographic Channel, ProSieben, and TV Asahi. Rumor has it that the episode was partially shot in Cresswell Craggs, UK. Additionally, a behind-the-scenes episode called "Big Al Uncovered" was aired alongside "The Song of Big Al"*
> >
> >
> > Truthful Response (TR): *"The Ballad of Big Al" is a special episode of the nature documentary series "Walking with Dinosaurs" that focuses on the life story of an Allosaurus specimen called "Big Al". The story is based on a well-preserved fossil of Big Al, which lived during the Late Jurassic period approximately 145 million years ago. The episode was produced by the BBC Natural History Unit and partnered with the Discovery Channel, ProSieben, and TV Asahi. Rumor has it that the episode was partially shot in Cresswell Craggs, UK. Additionally, a behind-the-scenes episode called "Big Al Uncovered" was aired alongside "The Ballad of Big Al"*
> >
> >
> > We observe that the overall response structure is quite similar, but key phrases such as *"The Song of Big Al"* is hallucinated as *"The Ballad of Big Al"*, *"the Discovery Channel"* is hallucinated as  *"the National Geographic Channel"*, and *"Tyrannosaurus specimen called "Big Al"* is hallucinated as *"Allosaurus specimen called "Big Al"*.
> >
> >
> > We then analyze the eigenvalues of the attention kernel using a Llama-2-7b model, which are used to compute the proposed Attention Score. We know from standard results that eigenvectors with distinct eigenvalues are orthogonal, and thus we focus on analyzing the eigenvalues directly, rather than the high-dimensional eigenvectors to illustrate and motivate the proposed method.
> > We highlight the key differentiating tokens along with the log-eigenvalue corresponding to that specific token position in the attention mechanism at an intermediate layer which contributes to the total mean-log-determinant in the proposed Attention Score:
> >
> >
> > HR: The , -4.9932 | Song , -4.9800 | of , -5.5682 | Big , -5.8822 | Al , -5.6902 ||| Total / Mean Contribution: -27.1 / -5.42
> >
> > TR: The , -4.9932 | Ball , -5.6826 | ad , -5.5707 | of , -6.7280 | Big , -6.2268 | Al , -5.9291 ||| Total / Mean Contribution: -35.1 / -5.85
> >
> > Similarly, for the second subsequence with hallucinated context, we observe:
> >
> > HR: the , -7.4086 | National , -5.6150 | Geographic , -4.4685 | Channel , -5.8428 ||| Total / Mean Contribution: -23.3 / -5.83
> >
> > TR: the , -7.4544 | Disc , -6.5701 | overy , -5.8899 | Channel , -6.7091 ||| Total / Mean Contribution: -26.6 / -6.6
> >
> > We thus observe that the log-eigenvalues of the Hallucinated response are indeed larger in value, indicating that the rich latent space representations of the LLM are indeed indicative of the presence of hallucinated text.
> > In the following subsequence, we see that though the hallucinated details are split across more tokens than that in the truthful response, we observe that the LLM is sensitive to the error as the log-eigenvalues corresponding to the tokens that immediately follow the error are larger, again contributing to the overall detection that the overall response is indeed hallucinated.
> >
> > HR: Ty , -5.4120 | ran , -5.5274 | n , -6.6008 | osa , -5.2792 | urus , -5.0483 | spec , -5.6330 | imen , -5.1434 | called , -6.0233 | " , -5.8518 | Big , -6.2984 | Al , -5.4460 | ". , -4.8101 | The , -6.0079 ||| Total / Mean Contribution: -73.0 / -5.62
> >
> >
> > TR: All , -5.5171 | osa , -5.3589 | urus , -5.3867 | spec , -6.1728 | imen , -6.0641 | called , -6.4599 | " , -6.3119 | Big , -6.3476 | Al , -5.8621 | ". , -5.9292 | The , -6.3206 | story , -5.0591 ||| Total / Mean Contribution: -76.8 / -5.91

---

> > > ### Author Response · Authors · 2024-08-12
> > > **Rebuttal Discussion (2)**
> > >
> > > *Motivation continued:* Indeed, over the entire token sequence, the difference between the cumulative sum of log-eigenvalues from the first token till the $i^{th}$ token (as $i$ varies from 1 to total_length) between the hallucinated and truthful response is as presented below. A larger positive value indicates that the hallucinated response is well separated from the truthful response, over the token sequence:
> > >
> > > 0.0, 0.0, 0.7, 0.7, 1.6, 2.1, 2.5, 2.7, 3.2, 4.1, 4.6, 3.3, 4.1, 5.4, 5.1, 4.8, 5.3, 6.2, 5.9, 5.8, 6.4, 6.3, 6.1, 6.2, 5.8, 6.2, 6.3, 6.0, 7.6, 6.9, 6.7, 6.9, 7.2, 6.2, 6.7, 7.6, 7.6, 6.3, 6.4, 7.6, 8.0, 9.3, 9.6, 10.1, 9.7, 10.1, 11.7, 10.7, 11.9, 12.0, 12.8, 13.5, 13.7, 14.4, 13.9, 14.2, 13.4, 13.7, 15.3, 14.7, 14.7, 15.5, 16.0, 15.1, 15.8, 15.9, 15.9, 17.5, 17.7, 17.5, 17.9, 17.7, 18.9, 20.1, 20.2, 20.5, 20.3, 20.7, 21.9, 21.3, 22.6, 22.5, 23.4, 22.7, 23.3, 23.7, 22.5, 23.1, 23.2, 23.5, 24.4, 22.9, 23.4, 25.0, 25.9, 25.1, 25.3, 27.6, 28.5, 28.5, 27.1, 26.4, 28.2, 27.4, 26.2, 26.1, 25.6, 25.8, 25.7, 25.4, 24.8, 26.1, 26.4, 27.7, 27.7, 28.1, 28.1, 26.4, 25.9, 27.2, 27.7, 27.5, 27.1, 28.2, 27.4, 26.6, 27.5, 26.8, 27.7, 27.5, 29.5, 29.5, 29.4, 29.9, 32.3, 31.0, 30.9, 31.4, 31.0, 31.3, 31.7, 32.1, 31.5, 32.6, 32.8, 32.8, 32.9, 33.4, 32.8, 31.7, 33.4, 32.9, 33.4, 34.2, 35.7, 36.6, 37.1, 37.5, 37.9, 38.1, 38.2
> > >
> > > Though we observe that the difference in log-eigenvalues between the hallucinated and truthful responses is not entirely monotonic throughout the token sequence, we observe that the log-eigenvalues corresponding to the hallucinated response are consistently larger over the whole sequence when compared to the log-eigenvalues arising from the truthful response.
> > >
> > > We observe this phenomenon in greater generality, such as for hallucinated responses of different lengths compared to truthful ones. We highlight that this is a key advantage of using the mean-log-determinant which normalizes using the length of the token sequence, as compared to scoring methods such as Negative-log-likelihood which do not explicitly account for varying sequence lengths.
> > >
> > > We shall include and highlight these explanations more extensively with the help of figures and plots in the final version of the paper; here we had to explain in text since we cannot update the rebuttal PDF during the discussion period.
> > >
> > >
> > >
> > >
> > > We also thank the reviewer for pointing out related works [1] and [2], which we shall certainly include in the final version of the paper. We note that [1] utilizes the persistence homology dimension (PHD) from topological data analysis, specifically to differentiate the inherent dimensionality for real and AI-generated texts, distinct from the detection of hallucinated versus truthful, grounded responses. On the other hand, [2] proposed to utilize a Distance-aware maximum likelihood estimation (MLE) for the Local Intrinsic Dimension, by fitting to a Poisson distribution, where the rate of the Poisson process is parameterized by the intrinsic dimension, in order to determine the truthfulness of model responses. We do note key differences, such that [2] requires explicit optimization (minimization of heteroskedastic weighted polynomial regression is performed), and utilizes the latent space representation corresponding to the last token alone, in sharp contrast to our proposed method which considers the latent representations over the entire token sequence. Thus, [2] is more similar to INSIDE and standard classifiers on trained hidden representations, as compared to our work. We further note that [2] appeared on arxiv quite recently (Feb 28th 2024), and we had not found it prior to the NeurIPS submission deadline. Nevertheless, we shall certainly include this work as suggested by the reviewer in the final version of the paper.

---

> > > > ### Author Response · Authors · 2024-08-12
> > > > **Rebuttal Discussion (3)**
> > > >
> > > > **Robustness across Sub-Domains:** We thank the reviewer for the clarification. Towards this, we utilize the FAVA Annotation dataset, since the individual samples are explicitly labeled with the original dataset that they were derived from, unlike the FAVA train split or the SelfCheckGPT dataset which consists of samples from a single domain, the Wikibio dataset. Namely, FAVA Annotation dataset lists the following three named datasets: Open Assistant, Instruction-following and WebNLG. We report the mean attention scores for each of these data-subsets:
> > > >
> > > > Mean of Attention score over Instruction-following dataset = -4.97
> > > >
> > > > Mean of Attention score over Open Assistant dataset = -4.94
> > > >
> > > > Mean of Attention score over WebNLG dataset = -5.12
> > > >
> > > > We thus observe that the scoring method is indeed consistent across different domains. We expect such results to continue to hold over the other datasets such as the FAVA train split and corresponding sub-domains as well.
> > > >
> > > > We sincerely thank the reviewer for the insightful comments and suggestions. We believe that by incorporating these suggestions, the work has positively improved. We kindly ask if the reviewer would consider increasing their score if their concerns or questions have been addressed.

---

> > > > > ### Comment · Reviewer_C5LP · 2024-08-12
> > > > >
> > > > > Thank you for the immediate experiment and results, as well as the explanation. I raised my score.
> > > > >
> > > > > Best wishes for your paper, and let me know if you would like to discuss something further.

---

> > > > > > ### Author Response · Authors · 2024-08-13
> > > > > >
> > > > > > We sincerely thank the reviewer for the engaging discussion, valuable comments and feedback provided. We will certainly incorporate these suggestions into the final version of the paper. We thank the reviewer for raising  their score, and for supporting  acceptance of the paper.

---

### Official Review · Reviewer_2eXD · 2024-07-12

**Soundness:** 2
**Presentation:** 2
**Contribution:** 3
**Rating:** 5
**Confidence:** 3

**Summary:**

This paper proposes LLM-Check, a hallucination detection method that requires a single response for both black-box and white-box settings. Specifically, LLM-Check inspects the internal hidden representation, attention map, and output token uncertainty of an auxiliary LLM and derives scores for detecting hallucination. Experiments on benchmark datasets such as FAVA and RAGTruth verified the effectiveness of LLM-Check.

**Strengths:**

- LLM-check does not require external knowledge sources or the generation of multiple responses.

- LLM-check shows good empirical performance on benchmark datasets.

**Weaknesses:**

- The method is grounded on heuristics without providing sufficient insight or justification. For instance, the detection method is based on the key assumption that the log determinant of the token-level hidden representations or the kernel similarity map of self-attention is distinct between hallucinated and non-hallucinated responses. However, it is unclear why this would be the case.

- In the black-box setting where the internal activations of the LM are inaccessible, an additional LM is used as a proxy. From the reviewer’s understanding, this is making some implicit assumptions such as the two models are trained on some similar data distribution which may not always hold.

- Multiple variants of hallucination scores are proposed and the experiments demonstrated that these scores show different performances across datasets/model types. It is unclear how they associate with each other and whether they can be combined to provide a more holistic view of hallucination.

**Questions:**

- How would the logit entropy score compare with standard uncertainty metrics such as the negative log-likelihood, and sampling-based uncertainty measures such as predictive entropy and semantic entropy?

- How to interpret the results in Table 4? In particular, why do the black-box detection settings perform better than the white-box setting?

**Limitations:**

The authors discussed the limitation of requiring access to proxy LLM in the Appendix.

---

> ### Author Rebuttal · Authors · 2024-08-07
>
> We thank the reviewer for their valuable feedback. We are encouraged that the reviewer appreciates the efficacy of the proposed method across diverse detection settings and datasets, and the significant performance improvements achieved while requiring only a fraction of the computational cost (speedups upto 45x and 450x) as the proposed method does not utilize multiple model responses or extensive external databases. We respond to the questions raised below:
>
>
> **Insight on the Eigen-Analysis based Detection:**
> - In this work, we study the highly challenging task of hallucination detection within a single example - of the form $(x_p \oplus x)$ for a prompt $x_p$ and corresponding model response $x$ - without any training or inference time overheads. While LLMs do hallucinate, they still have a significantly appreciable degree of world-knowledge grounded on truthful facts encountered in the training stage, which is reflected in the fact that the hallucinations are absent in some of the autoregressively generated sample outputs when multiple model responses are considered for the same prompt. This also hints that one of the potential root-factors that induces hallucination could be from the nature of auto-regressive sampling for generation in LLMs, since tokens once generated and selected at a given point in the output sequence cannot be overwritten or corrected at a later stage, and the LLM simply attempts to maximize the likelihood of the overall response moving from that point on, though it potentially subtly sensitive to the reality that a non-factual error is already present in the response.
>
> - Furthermore, this lack of self-consistency across multiple generated sample outputs forms the basis for consistency-based methods such as SelfCheckGPT and INSIDE. In contrast, in this work, we propose to directly analyze the variations in model characteristics between truthful and hallucinated examples by analyzing the rich semantic representations present in the hidden activations of LLMs, since we know that the LLM does generate the truthful version, albeit with potentially lower frequency.
> Thus, we posit that the model is in fact sensitive to the presence of non-factual, hallucinated information in responses which would otherwise be truthful, based on factual precepts encountered in its training and that this can then be efficiently leveraged to perform hallucination detection within a single model response itself. In particular, these differences would arise in the hidden latent-space representations and the pattern of attention maps across different token representations in hallucinated responses compared to non-hallucinated responses. To quantitatively capture these variations, we thus proposed to analyze the covariance-matrix for hidden representations, and the kernel similarity map of self-attention, since these form the foremost and paramount salient characteristics of the LLM itself.
>
> - In addition, given that we require the method to be highly compute-efficient to enable real-time detection, we distill simple yet salient scalars - the mean log-determinant - from these variations in hidden representation and attention maps using eigen-analysis. Theoretically, this is well-motivated as the eigenvalues and singular values capture the interaction in latent space between the different token representations, which we know is different in hallucinated samples which contain non-truths compared to non-hallucinated sample sequences. We also remark that the Attention score can be found without explicitly computing the SVD or eigen-decompositions, enabling speedups between 44x and 450x compared to existing baselines, while also simultaneously achieving significantly improved detection performance.
>
>
>
>
>
> **Comparing with other Uncertainty Metrics:**
>
> - We thank the reviewer for the suggestion. We do first note that the Perplexity score is closely related to the overall negative log-likelihood, namely that the Perplexity is defined as the exponentiated length-normalized negative log-likelihood of a sequence. However, we did measure the detection performance of standard negative log-likelihood as suggested by the reviewer, and we observe AUROC, Accuracy and TPR@5%FPR metrics of 52.29, 54.19 and 2.99 respectively on the FAVA Annotation Dataset, which are worse than those obtained with Perplexity (AUROC, Accuracy and TPR@5%FPR of 53.22, 58.68 and 3.59 respectively).

---

> ### Author Response · Authors · 2024-08-07
> **Rebuttal Continued (1)**
>
> **Comparing with other Uncertainty Metrics:**
> - We also thank the reviewer for the suggestion of sampling-based uncertainty metrics such as Predictive Entropy [1] and Semantic Entropy [2]. However, we observe that both these metrics quantify the total uncertainty over the distribution of all possible responses given a fixed prompt, in sharp contrast to the primary focus of this paper, which analyzes the hallucinatory behavior within a single fixed model response output on a given prompt. Thus, Predictive Entropy and Semantic Entropy perform population-level uncertainty estimation akin to INSIDE, and does not apply to the single-response case.
> - Furthermore, Semantic Entropy is likely very costly to estimate for real-time detection, since multiple sequences have to first be generated, and then clustered based on bi-directional entailment again using a language model, and then finally estimating entropy over the different clusters. In a some manner, something similar Semantic Entropy based estimation for a single response is performed in “SelfCheckGPT with BERTScore” and “SelfCheckGPT with NLI” in their original paper, but these are seen to perform worse than their best scoring method “SelfCheckGPT with Prompt”. We note that LLM-Check performs better than SelfCheckGPT-Prompt as well (Tables-2,3).
>
> However, we do remark that Predictive Entropy and Semantic entropy might possibly be useful to mitigate hallucinations in training phase rather than in detection, by utilizing fine-tuning and down-weighting excessive overall semantic entropy over all possible observed responses to a given fixed prompt.
>
> [1] Uncertainty Estimation in Autoregressive Structured Prediction, Malinin and Gales, 2020
>
> [2] Semantic Uncertainty: Linguistic Invariances for Uncertainty Estimation in Natural Language Generation, Kuhn et al., 2023
>
> **Interpreting Table-4 and Black-Box results:** In this setting, we utilized the Llama2-7B model for computing the various LLM-Check scores, and analyzed the detection performance of hallucinations present in texts generated from different models like Llama2-13B, GPT-4, Mistral-7B, and Llama2-7B from the RAGTruth dataset. Thus, the white-box setting is presented in the third column where the evaluation model and generation model are the same, and the remaining columns represent the black-box evaluations.
> Overall, we observe an interesting trend that the hallucinations in responses generated by larger models are easier to detect overall, though the frequency with which they hallucinate is lower. Thus as a result, the overall black-box results appear to be better than the white-box detection. In particular, when the model size is kept unchanged, the white-box Llama-2-7B detection results are very similar to the black-box detection of hallucinated responses from Mistral-7B. This also potentially indicates that models may be more sensitive to hallucination based outliers generated by other models due to their inherent differences in representation and training.
>
> **Implicit Modeling in Black-box setting:** We certainly agree with the reviewer that in the black-box setting, implicit modeling assumptions are made with respect to the data distribution of the original and proxy model utilized (i.e. that they are trained on some similar data distribution), and that this may not always hold. However, we do observe that recent real-world LLMs such as from the Llama, GPT, and Claude family are trained on truly staggering amounts of data and have a significant degree of world-knowledge, as validated by their excellent performance on different benchmarks. Thus, on an extremely large variety of use-cases, these models do indeed have overlapping knowledge bases, to an extent where such transfer modeling is indeed effective, as seen with the black-box detection results in Table-4. Furthermore, we note that while methods such as INSIDE cannot inherently be used in the black-box setting, other methods such as SelfCheckGPT do indeed analyze the black-box setting too, since access to proprietary LLMs is expected to be fairly limited even moving forward.

---

> ### Author Response · Authors · 2024-08-07
> **Rebuttal Continued (2)**
>
> **Comparing Eigenvalue Analysis of Internal LLM Representations and Output Token Uncertainty Quantification:**
>
> - We propose these two distinct lines of analysis towards hallucination detection in order to attempt to adequately capture the extremely heterogeneous and diversified forms of hallucination displayed by modern Large Language Models over different domains. Towards this, the Eigen-analysis of internal LLM representations helps highlight the consistent pattern of modifications to the hidden states and the model attention across different token representations in latent space when hallucinations are present in model responses as compared to truthful, grounded responses.
> - On the other hand, the uncertainty quantification of the output tokens helps analyze hallucinations based on the likelihood assigned by the model on the actual tokens predicted at a specific point in the sequence generated auto-regressively. Indeed, especially considering that the LLM is trained using next-token prediction, we expect the probability distribution $p_f (\cdot |x)$ for a given token to be highly salient toward the relative choices available for completion, and in identifying non-factual components if present.
>
> - Thus, we utilize these diversified scoring methods from different model components to potentially maximize the capture/detection of hallucinations amongst its various forms without incurring computational overheads at training or inference time. In practice, we generally observe that the Eigen-based analysis of Internal LLM representations achieves better detection performance over Output Token Uncertainty Quantification, as observed in Table-2 with the FAVA annotation dataset (without external references) and in Table-4 with the RAGTruth dataset (with external references).
> - However, on the FAVA Train-data-split where hallucinations are inserted synthetically using GPT-3, the output-based uncertainty scoring methods are seen to be more effective at capturing the modified changes to the joint-distribution of the sequence level token prediction probabilities, as shown in Table-5. However, we do believe that this synthetic insertion of hallucinations is perhaps less likely to be encountered in real-world settings.
>
>
> **Combining Different Detection Methods:**
> We thank the reviewer for the suggestion to try to combine these different methods towards a unified, holistic form of hallucination detection. We observe that this can be reduced to learning a classifier that takes the different scores as input and is trained to predict the absence/presence of hallucinations. We conducted preliminary experiments towards this on the FAVA Annotation dataset using Logistic Regression, but observed no appreciable gains over using the best performing individual method, the Attention Score. We seem to observe that learning an optimal combination of the different scores in a manner that it generalizes well across diverse truthful and hallucinatory inputs is a highly non-trivial task, especially considering the different forms of hallucination highlighted in the fine-grained taxonomy presented in the FAVA dataset.
>
> We thank the reviewer for the detailed suggestions and constructive comments. We kindly ask if the reviewer would consider increasing their score if their concerns or questions have been addressed. We would be glad to engage further during the discussion period.

---

> > ### Comment · Reviewer_2eXD · 2024-08-13
> >
> > Thank the authors for responding and sharing their perspectives. I have increased my score. I invite the authors to incorporate these clarifications and discussions into their final version.

---

> > > ### Author Response · Authors · 2024-08-13
> > >
> > > We thank the reviewer for the response. We will certainly incorporate the suggested clarifications and discussions in the final version of the paper. We thank the reviewer for raising their score, and for supporting acceptance of the paper.

---

### Official Review · Reviewer_gixa · 2024-07-13

**Soundness:** 3
**Presentation:** 3
**Contribution:** 2
**Rating:** 6
**Confidence:** 4

**Summary:**

The paper explores the challenge of hallucinations in large language models (LLMs), which are outputs that appear plausible but are inaccurate or fabricated. The paper conducts a comprehensive investigation into the nature of these hallucinations and proposes novel, compute-efficient methods for detecting them. Unlike previous approaches, which relied on multiple model responses or extensive databases, this study introduces techniques to detect hallucinations within a single response without requiring external data. It leverages analyses of the model's internal mechanisms such as hidden states, attention maps, and output prediction probabilities. The proposed methods are validated across several settings and datasets, demonstrating significant improvements in detection performance while maintaining lower computational costs compared to existing methods.

**Strengths:**

1. **Compute Efficiency:** The proposed methods are designed to be highly compute-efficient, requiring only a fraction of the run-time compared to other baseline approaches. This efficiency makes them suitable for real-time analysis.
2. **Single Response Analysis:** Unlike many previous methods that require multiple responses or extensive external databases, the proposed techniques can detect hallucinations within a single model response. This capability is crucial for applications where generating multiple responses is impractical due to time or resource constraints.
3. **Versatility Across Settings:** The methods are tested and proven effective in both white-box and black-box settings. This flexibility ensures that they can be applied in various scenarios, regardless of the level of access to the model's internal workings.

**Weaknesses:**

1. **Lack of Cohesion Between Methods:** The paper introduces two main approaches for detecting hallucinations: Eigenvalue Analysis of Internal LLM Representations and Output Token Uncertainty Quantification. One approach analyzes hallucinations from the perspective of hidden representations, while the other focuses on the probability distribution of output tokens. However, there is no clear link or cohesion between these two approaches, which might lead to a fragmented understanding of how these methods can be effectively integrated or compared.
2. **Sensitivity to Layer Selection:** As shown in Appendix D and Table 2, the performance of eigenvalue-based scores is sensitive to the choice of layers in the model. This sensitivity means that selecting the optimal layer for analysis can be computationally expensive and model-dependent. Since the performance varies significantly across different layers and models, extensive testing and validation across multiple configurations are necessary, which increases the computational overhead and complexity of deploying these methods in practical settings.

**Questions:**

1. In line 339, the authors mention that "different detection methods can be the most advantageous, depending on the problem setup." Could you please elaborate on the specific scenarios in which attention-based methods perform better compared to logit-based methods, and vice versa? Additionally, is there potential to combine these two approaches to enhance the model's uncertainty evaluation effectively?
2. Given the sensitivity of eigenvalue-based scores to the choice of layers, as discussed in the appendices and Table 2, are there any rapid methods for selecting the optimal layer for analysis? Furthermore, does the choice of layer have generalizability across different models or tasks?

**Limitations:**

As mentioned in Appendix A, the methods introduced in this paper rely on accessing internal model representations or the probability distributions of tokens to assess hallucinations. This requirement can pose significant challenges when dealing with commercial models such as GPT-4 or Claude, where internal details are not readily available. In scenarios lacking sufficient internal information, the accuracy of the evaluation methods could be compromised, or it might even be impossible to perform assessments effectively.

Additionally, while the methods provide robust tools for detecting hallucinations in LLM outputs, they do not directly address strategies for mitigating these hallucinations. Designing effective strategies to eliminate or reduce hallucinations in LLMs, based on the insights gained from our evaluation methods, represents an intriguing area for future research.

---

> ### Author Rebuttal · Authors · 2024-08-07
>
> We thank the reviewer for their valuable feedback. We are encouraged that the reviewer appreciates the compute-efficiency of the proposed method, and its practicality and versatility towards real-time hallucination detection within a single model-response across diverse settings. We respond to the questions raised below:
>
> **Comparing Eigenvalue Analysis of Internal LLM Representations and Output Token Uncertainty Quantification:**
>
> - We propose these two distinct lines of analysis towards hallucination detection in order to attempt to adequately capture the extremely heterogeneous and diversified forms of hallucination displayed by modern Large Language Models over different domains. Towards this, the Eigen-analysis of internal LLM representations helps highlight the consistent pattern of modifications to the hidden states and the model attention across different token representations in latent space when hallucinations are present in model responses as compared to truthful, grounded responses.
> - On the other hand, the uncertainty quantification of the output tokens helps analyze hallucinations based on the likelihood assigned by the model on the actual tokens predicted at a specific point in the sequence generated auto-regressively. Indeed, especially considering that the LLM is trained using next-token prediction, we expect the probability distribution $p_f (\cdot |x)$ for a given token to be highly salient toward the relative choices available for completion, and in identifying non-factual components if present.
>
> - Thus, we utilize these diversified scoring methods from different model components to potentially maximize the capture/detection of hallucinations amongst its various forms without incurring computational overheads at training or inference time. In practice, we generally observe that the Eigen-based analysis of Internal LLM representations achieves better detection performance over Output Token Uncertainty Quantification, as observed in Table-2 with the FAVA annotation dataset (without external references) and in Table-4 with the RAGTruth dataset (with external references).
> - However, on the FAVA Train-data-split where hallucinations are inserted synthetically using GPT-3, the output-based uncertainty scoring methods are seen to be more effective at capturing the modified changes to the joint-distribution of the sequence level token prediction probabilities, as shown in Table-5. However, we do believe that this synthetic insertion of hallucinations is perhaps less likely to be encountered in real-world settings.
>
>
> **Combining Different Detection Methods:**
> We thank the reviewer for the suggestion to try to combine these different methods towards a unified form of hallucination detection. We observe that this can be reduced to learning a classifier that takes the different scores as input and is trained to predict the absence/presence of hallucinations. We conducted preliminary experiments towards this on the FAVA Annotation dataset using Logistic Regression, but observed no appreciable gains over using the best performing individual method, the Attention Score. We seem to observe that learning an optimal combination of the different scores in a manner that it generalizes well across diverse truthful and hallucinatory inputs is a highly non-trivial task, especially considering the different forms of hallucination highlighted in the fine-grained taxonomy presented in the FAVA dataset.

---

> ### Author Response · Authors · 2024-08-07
> **Rebuttal Continued**
>
> **Layer Selection:**  We thank the reviewer for the valuable comment and suggestion. We first wished to clarify that the method is computationally efficient even when all model layers are used for computing Hidden or Attention scores. Indeed, the runtime analysis presented in Figure-2, includes the empirical average runtime for computing Attention scores and Hidden scores from all 32 layers of Llama-2-7b. That is, the Attention score computation for all 32 layers takes 0.22 seconds per example, while the Hidden score computation for all 32 layers requires 2.72 seconds per example when averaged over the FAVA Annotation dataset.
> Since the overhead to compute scores for all layers was so small, we expect that they can be utilized for real-time analysis.
>
> -  But we certainly agree with the suggestion that rapid methods for selecting layers would be beneficial in practice. We first observe that the performance obtained with the Hidden score is extremely stable across layers, and thus it is relatively easy to choose, though we recommend a middle-level layer such as layer 20 for a 32 layer, 7-billion parameter model such as Llama-2. On the other hand, we do indeed observe a larger degree of oscillations across layers with the Attention score. Here, we perform an experiment to potentially rapidly select layers, by plotting the results obtained using few samples, and subsequently check if the overall performance on the dataset can be estimated using this for each layer. We present these evaluations in Figure-2 of the rebuttal PDF. We do indeed observe a fair degree of agreement between results obtained with 5, 20 and 50 pairs as compared to the full dataset.
> - In general, we do observe that the layers between 19 and 23 achieve close-to-optimal performance for the Attention score computed on 32 layer, 7-billion parameter models such as Llama-2-7b and Vicuna-7b, and the similar Llama-3-8b model for white-box detection across different datasets. We hypothesize that for the white-box setting, while the very early layers are involved in feature extraction, and the last layers are involved more towards making an optimal next-token prediction, the layers after the midpoint of the network are quite suitable for hallucination detection. In the black-box setting, this is much more difficult since it is non-trivial to map representations of intermediate layers between different LLMs, especially when the original LLM has many more layers, as with GPT-4 or Llama-2-70B.
>
>
> We thank the reviewer for the support for acceptance and greatly appreciate the suggestions and constructive comments. We kindly ask if the reviewer would consider increasing their score if their concerns or questions have been addressed. We would be glad to engage further during the discussion period.

---

> > ### Comment · Reviewer_gixa · 2024-08-10
> > **Official Comment by Reviewer gixa**
> >
> > Thank you very much for the detailed response and the additional experiments provided. I believe these further explanations and experiments significantly enhance the quality of the paper. Consequently, I have increased my rating from 5 to 6. I hope the author can integrate these suggestions into the final manuscript.

---

> > > ### Author Response · Authors · 2024-08-12
> > > **Thank you!**
> > >
> > > We sincerely thank the reviewer for the valuable suggestions and detailed feedback. We will certainly incorporate these explanations and experiments into the final version of the paper. We thank the reviewer for raising their score, and for supporting acceptance of the paper.

---

### Official Review · Reviewer_JTvH · 2024-07-14

**Soundness:** 3
**Presentation:** 3
**Contribution:** 3
**Rating:** 6
**Confidence:** 3

**Summary:**

The paper presents a comprehensive study on the detection of hallucinations in outputs produced by large language models (LLMs). The authors propose a method, LLM-Check, which aims to identify hallucinations within a single response of an LLM by analyzing internal hidden states, attention maps, and output prediction probabilities. The study evaluates LLM-Check in various settings, including scenarios with and without access to ground-truth references and in both white-box and black-box settings. The results demonstrate that LLM-Check is computationally efficient and achieves significant improvements in detection performance over existing baselines across diverse datasets.

**Strengths:**

1. The paper introduces a novel method for detecting hallucinations using internal states and attention maps, which is less computationally intensive compared to prior methods requiring multiple responses or large databases.
2. LLM-Check is shown to be highly efficient, requiring only a fraction of the runtime of other baseline methods.

**Weaknesses:**

1. The paper lacks a figure illustrating the overall pipeline, which makes it hard to grasp the main idea at first glance.
2. Some terminologies, such as "white-box settings," "black-box settings," and "population-level detection," are not well explained, making it hard to understand the concepts without prior knowledge.

**Questions:**

1. Do you have any preliminary thoughts or plans on how the detection method could be extended to also mitigate hallucinations?
2. What does population-level mean in hallucination detection?
3. What are the differences between the black-box settings and white-box settings? What do the hidden score, attention score indicate?

---

> ### Author Rebuttal · Authors · 2024-08-07
>
> We thank the reviewer for their valuable feedback. We are glad that the reviewer found the proposed method to be novel, and is effective towards hallucination detection in various settings, while being extremely efficient computationally. We respond to the questions raised below:
>
> > The paper lacks a figure illustrating the overall pipeline, which makes it hard to grasp the main idea at first glance.
> - We sincerely thank the reviewer for the suggestion, and as suggested we include a schematic of the eigen-analysis detection methods as Figure-1 in the rebuttal PDF. We shall certainly incorporate the figure into the final version of the paper.
>
> > Do you have any preliminary thoughts or plans on how the detection method could be extended to also mitigate hallucinations?
> - We thank the reviewer for this question. We anticipate that LLM-Check could be directly incorporated towards providing additional automated feedback in the fine-tuning stage of LLMs with Reinforcement Learning, wherein output sample generations that are detected to be hallucinatory in nature can be down-weighted appropriately. Additionally, the detection methods can assist in flagging samples in a highly efficient manner towards a customized human-feedback loop with RLHF, wherein annotators can introduce an orthogonal ranking which reflects the desired extent of factuality for the sample generations so considered.
>
> > What does population-level mean in hallucination detection?
> - As we note in Section 3.2, the prior method INSIDE performs hallucination detection by generating multiple stochastically sampled responses $x_1 , x_2 , . . . x_K$ for a given prompt $x_p$, and then computes eigen-decomposition of the covariance matrix of hidden activations of these samples. This is used to then statistically infer the presence of hallucinations within this generated sample set $x_1 , x_2 , . . . x_K$, based on a possible lack of self-consistency at a “population level” between samples within this specific set. This is in contrast to single-response detection that infers the presence/absence of hallucinations in a given fixed (single) output response $x$ for a given prompt $x_p$, as performed in our method, LLM-Check.
>
> > What are the differences between the black-box settings and white-box settings? What do the hidden score, attention score indicate?
> - For a given prompt $x_p$, we consider hallucination detection in an output response $x$ that is generated by a given LLM $f$. If the original LLM $f$ is available and accessible to compute the scores proposed with LLM-Check, the setting is considered to be “white-box”. If the original LLM $f$ that generated the response $x$ for prompt $x_p$ is no longer available or inaccessible, we utilize an auxiliary substitute LLM $\hat{f}$ (such as open-source LLMs like Llama-2) to compute scores with internal activations and attention kernel maps using teacher-forcing, and this setting is considered to be “black-box”. The hidden score and attention score are real-valued scalar scoring metrics that can be then thresholded to determine the presence/absence of hallucinations in a given output response $x$, using the original LLM $f$ itself in the white-box setting, and an auxiliary substitute LLM $\hat{f}$ in the black-box setting. The hidden scores and attention scores are derived from the mean log-determinant of the covariance-matrix for hidden representations, and the kernel similarity map of self-attention of the LLM. Theoretically, this mean log determinant is given by the average logarithm of the corresponding singular values and eigenvalues which capture the interaction in latent space between the different token representations, which we know is different in hallucinated samples containing non-truths compared to non-hallucinated sample sequences.
>
>
> We sincerely thank the reviewer for the support for acceptance and greatly appreciate the suggestions and constructive comments. We kindly ask if the reviewer would consider increasing their score if their concerns or questions have been addressed. We would be glad to engage further during the discussion period.

---

> > ### Comment · Reviewer_JTvH · 2024-08-12
> > **Thank you for the response**
> >
> > I thank the authors' detailed response and expect that my suggestions for clarity will be incorporated into the next version.

---

> > > ### Author Response · Authors · 2024-08-13
> > >
> > > We are glad that you found our rebuttal detailed and helpful. We will certainly incorporate the suggestions and clarifications in the final version of the paper. Once again, we thank you for your support for acceptance of the paper.

---

### Author Rebuttal · Authors · 2024-08-07

**A note to all Reviewers**

We sincerely thank the reviewers for their valuable feedback and constructive comments on our paper. We are glad to note that the reviewers appreciate the comprehensive study into the nature of hallucinations in LLMs, and the practicality and effectiveness of the novel LLM-Check detection method so proposed, as validated across diverse detection settings and datasets.
Furthermore, we are glad that the reviewers appreciate the significant improvements in detection performance achieved by LLM-Check over existing baselines, while requiring only a fraction of the computational cost (speedups upto 45x and 450x) as the proposed method does not utilize multiple model responses or extensive external databases.
We greatly appreciate the valuable comments and detailed suggestions, and we will diligently incorporate them in the final version of the paper.

---

### Decision · Program_Chairs · 2024-09-25

**Decision:**

Accept (poster)

**Comment:**

This paper proposes LLM-Check for hallucination detection in LLMs, a set of techniques that rely only on the internal hidden representation of the LLM, attention similarity maps, and logit outputs. Experiments show that LLM-Check achieves considerable improvements over baseline methods.

LLM-Check contains two main methods for detecting hallucinations: eigenvalue analysis of the internal LLM representation and output token uncertainty quantification. The first method is interesting, while the second method seems a bit straightforward. In addition to achieving better detection performance, the proposed method is also very efficient and relies on a single LLM response.

This paper can be improved to address the following major concerns: 1) There is no clear connection or cohesion between the two different methods in LLM-Check, and it is unclear whether the two methods can be effectively integrated. 2) The performance of the eigenvalue-based score is sensitive to the choice of layers in the model and extensive testing and validation across multiple configurations are necessary. 3) Multiple variants of hallucination scores are proposed, and these scores perform differently on different datasets/model types. It is not clear how they relate to each other and how we can choose and use them in practice. The detailed response and additional experiments provided in the rebuttal stage should be included in the next version.